# Probing sub-5 Ångstrom micropores in carbon for precise light olefin/paraffin separation

Shengjun Du [1,9], Jiawu Huang [1,9], Matthew R. Ryder [2], Luke L. Daemen[3], Cuiting Yang[1], Hongjun Zhang [4], Panchao Yin[5], Yuyan Lai[5], Jing Xiao [1]✉, Sheng Dai [6,7]✉ & Banglin Chen [8]✉

Olefin/paraffin separation is an important but challenging and energy-intensive process in petrochemical industry. The realization of carbons with size-exclusion capability is highly desirable but rarely reported. Herein, we report polydopamine-derived carbons (PDA-Cx, where x refers to the pyrolysis temperature) with tailorable sub-5 Å micropore orifices together with larger microvoids by one-step pyrolysis. The sub-5 Å micropore orifices centered at 4.1–4.3 Å in PDA-C800 and 3.7–4.0 Å in PDA-C900 allow the entry of olefins while entirely excluding their paraffin counterparts, performing a precise cut-off to discriminate olefin/paraffin with sub-angstrom discrepancy. The larger voids enable high $C_2H_4$ and $C_3H_6$ capacities of 2.25 and 1.98 mmol g$^{-1}$ under ambient conditions, respectively. Breakthrough experiments confirm that a one-step adsorption-desorption process can obtain high-purity olefins. Inelastic neutron scattering further reveals the host–guest interaction of adsorbed $C_2H_4$ and $C_3H_6$ molecules in PDA-Cx. This study opens an avenue to exploit the sub-5 Å micropores in carbon and their desirable size-exclusion effect.

Light olefins, especially ethylene ($C_2H_4$) and propylene ($C_3H_6$), are crucial chemical feedstocks in petrochemical industries for the broad production of polyethylene, polypropylene, and highly value-added products[1–4]. The world demand for $C_2H_4$ and $C_3H_6$ in 2022 exceeds 210 and 140 million metric tons, respectively. Steam cracking is the major industrial approach for olefin production, in which the paraffins are inevitably entrained as the byproducts[5,6]. The separation of olefins from their paraffin counterparts is thus an essential process to achieve high-purity olefins[7–9], and has been listed as one of the seven major chemical separations to change the world[10]. Currently, industrial separation of olefin/paraffin mainly relies on energy-intensive cryogenic distillation (number of trays >100, pressure >20 bar, temperature <258 K), which accounts for approximately 0.3% of the total global energy consumption[11]. Driven by the world's energy footprint, adsorptive separation has become an attractive and reliable alternative due to its high efficiency, energy-saving, and flexible operation[12–14].

The keystone for adsorptive separation is the development of advanced physisorbents with excellent separation performance and scalability[15,16]. Porous carbon materials have demonstrated great prospects and broad application in gas separation due to their rich

[1]School of Chemistry and Chemical Engineering, South China University of Technology, Guangzhou, China. [2]Materials Science and Technology Division, Oak Ridge National Laboratory, Oak Ridge, TN, USA. [3]Neutron Scattering Division, Oak Ridge National Laboratory, Oak Ridge, TN, USA. [4]State Key Laboratory of Particle Detection and Electronics, University of Science and Technology of China, Hefei, China. [5]State Key Laboratory of Luminescent Materials and Devices, School of Molecular Science and Engineering, South China University of Technology, Guangzhou, China. [6]Department of Chemistry, Institute for Advanced Materials and Manufacturing, University of Tennessee, Knoxville, TN, USA. [7]Chemical Sciences Division, Oak Ridge National Laboratory, Oak Ridge, TN, USA. [8]Fujian Provincial Key Laboratory of Polymer Materials, College of Chemistry & Materials Science, Fujian Normal University, Fuzhou, Fujian, China. [9]These authors contributed equally: Shengjun Du, Jiawu Huang. ✉e-mail: cejingxiao@scut.edu.cn; dais@ornl.gov; banglin.chen@fjnu.edu.cn

porosity, excellent stability, and low cost[17–19]. However, regarding the olefin/paraffin separation, the conventional equilibrium separation based on favorable enthalpic interaction toward olefins or paraffins cannot confer carbon materials with desirable selectivity due to their similar physicochemical properties[20]. In principle, to maximize the separation factor, the ideal physisorbents should have narrower sub-nanometer micropores to match the guest molecule size, thus taking up smaller olefins while completely excluding larger paraffin counterparts by precise regulation of pore sizes or geometries[15,21]. However, the major barrier is the broadly distributed pore size of carbonaceous materials, ranging from sub-nanometer to micrometer scale, due to the random arrangement of carbonaceous nanodomains with uncontrollable defects[22]. Such broad pore size distributions (PSD) inevitably cause the co-adsorption of both olefins and paraffins, thus poor selectivity at a low uptake ratio[23,24].

In recent years, research endeavors have been devoted to designing carbon materials with tailorable porosity, especially with favorable ultrahigh surface area and pore volume for gas adsorption and storage[25,26], and only a few works reported carbons with suitable small micropores matching olefins over the slightly larger paraffin counterparts for selective separation. The carbon of C-CDMOF-2-700 derived from a metal-organic framework (MOF) was recently reported to enable the separation of $C_3H_6$ and $C_3H_8$ via a size-sieving effect[27]. Still, to our best knowledge, it is rather challenging to tailor the micropores (or micropore orifices) in carbon to a lower size range to distinguish $C_2H_4$ and $C_2H_6$ at sub-angstrom precision. Meanwhile, such small micropores in carbon can be readily tuned to recognize $C_3H_6$ over the $C_3H_8$ counterpart. Furthermore, the conventional single gas probe technique such as $N_2$ at 77 K and Ar at 87 K, mainly detects the larger micropores beyond 5 Å[28], thereby the contribution from small micropore below 5 Å in carbons can often be veiled and underestimated. As a complement, multiple gas probe molecules could provide a more comprehensive assessment[29], with cautiously chosen groups of probe molecules. In addition, unlike crystalline materials

with precise crystallographic data as an alternative, carbon materials with amorphous structures cannot generalize such an approach. The resolution of small micropores below 5 Å and their manipulation for important industrial separations is worthy of investigation and may open an avenue in the highly selective separation of olefin/paraffin at similar sizes on carbons.

Herein, we move toward an insight into specific pore-engineering on small micropore orifices below 5 Å in carbons, which realizes precise molecular recognition of $C_2H_4/C_2H_6$ and $C_3H_6/C_3H_8$ based on a size-exclusion effect. The PDA-Cx carbons were synthesized by the self-assembling of dopamine (DA) followed by pyrolysis of assembled polydopamine (PDA) with π-electron-rich backbones stacked along the z axis[30,31] (Fig. 1a). Being recognized as a non-porous solid by $N_2$ adsorption at 77 K, PDA-Cx possesses a high surface area up to 400 m² g⁻¹ by $CO_2$ adsorption at 195 K. Small sub-5 Å micropore orifices were resolved comprehensively by a series of gas probe molecules ranging from 3.3 to 5.0 Å at sub-angstrom size discrepancies. The PDA-C800 and PDA-C900 are sizing narrowly in the range of 4.1–4.3 Å and 3.7–4.0 Å, respectively. Such refined micropore orifices matched well with the pore size required for size-sieving separation of $C_2$- and $C_3$-olefin/paraffin pairs (Fig. 1b). Hence, PDA-Cx not only takes up high amounts of $C_2H_4$ (2.25 mmol g⁻¹) or $C_3H_6$ (1.98 mmol g⁻¹) at 298 K and 1.0 bar but also excludes $C_2H_6$ and $C_3H_8$ counterparts. Inelastic neutron scattering (INS) further examines host-guest interaction in carbons.

## Results
### Thermal-regulated structural evolution
The thermal stability of the polymer determines its successful transition to the carbonaceous structure during pyrolysis. TG analysis showed that even heated up to 900 °C for PDA polymer, 53.9% of the residues remained (Supplementary Fig. 1), much higher than many other polymer precursors and classical biomaterials[7,32]. Such excellent thermal stability contributes to the good preservation of sphere-

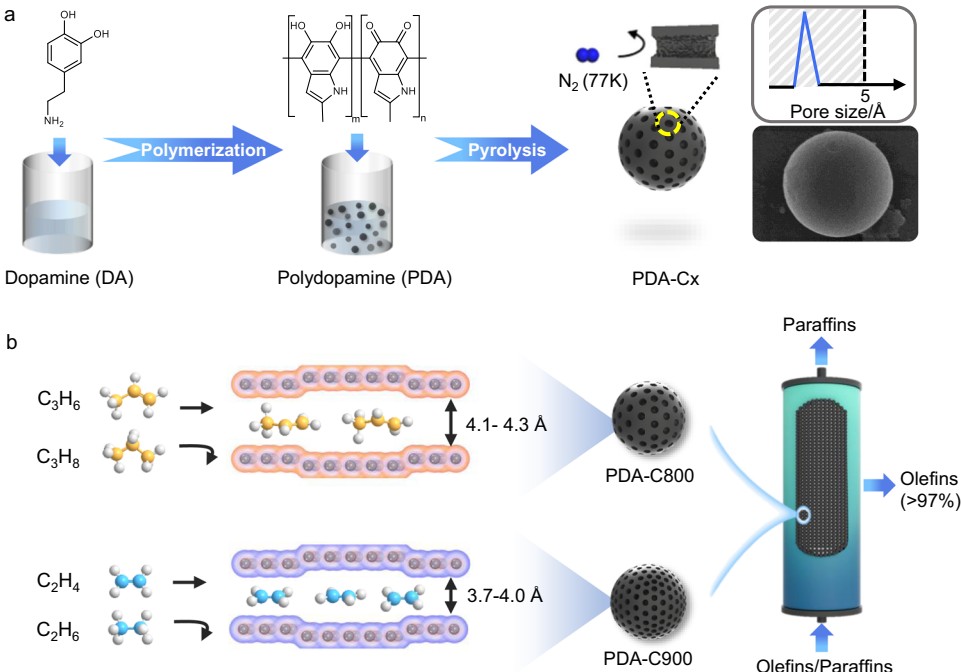

**Fig. 1 | Formation process and gas sorption properties of PDA-Cx. a** Schematic illustration of the synthetic procedure. Dopamine undergoes self-assembly to prepare polydopamine firstly and subsequent pyrolysis to synthesize PDA-Cx with narrowly distributed sub-5 Å micropore orifices; **b** Schematic illustrations of a side view of the pore system. The small micropore orifices in PDA-C800 (4.1–4.3 Å) and PDA-C900 (3.7–4.0 Å) were probed by a series of angstrom-sized gas probes ranging from 3.3 to 5.0 Å; PDA-C800 admits $C_3H_6$ (4.0 Å) while rejects $C_3H_8$ (4.3 Å) completely and PDA-C900 admits $C_2H_4$ (3.7 Å) while rejects $C_2H_6$ (4.1 Å) completely; such refined small micropores enable the production of high-purity olefins (>97%) through a one-step adsorption-desorption process.

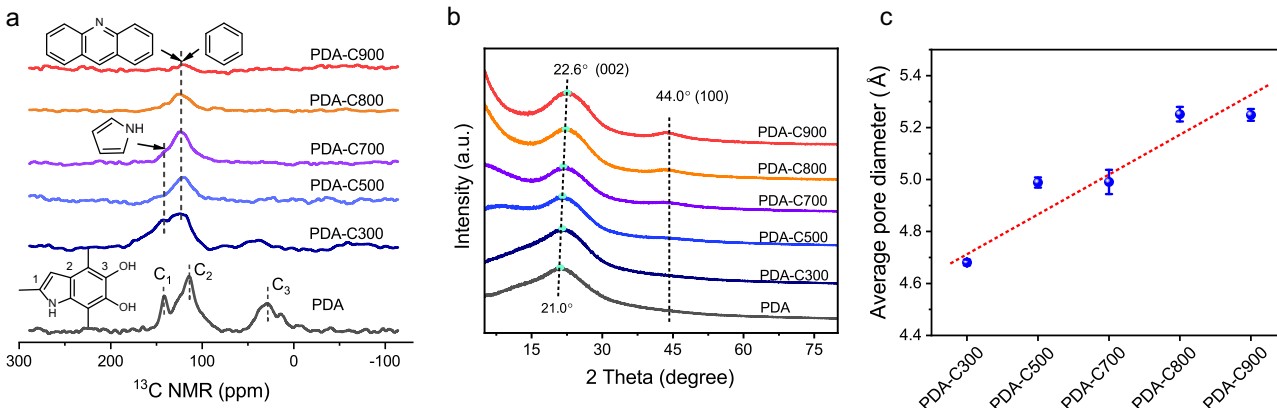

**Fig. 2 | Characterization of PDA and PDA-Cx. a** $^{13}$C CP MAS NMR experiments recorded on PDA and PDA-Cx; **b** PXRD patterns of PDA and PDA-Cx. The dotted lines mark the (002) and (100) peaks; **c** The average pore size of PDA-Cx derived from the PALS. The dashed line is a linear data fit as a visual guide. The error bars represent the standard deviations based on the nonlinear least squares fitting process carried out using the LTv9 program.

shaped morphology (Supplementary Fig. 2). The homogeneous elemental distribution of C, O, and N were found throughout the PDA in the EDS mapping, indicating a remarkable heteroatom-doping structure (Supplementary Fig. 3). To track the structural change of heteroatom bonding nature during pyrolysis, X-ray photoelectron spectra (XPS) was utilized, which indicates the transformation from pyrrolic-N to pyridinic-N and thermally stable graphitic-N, existing at edges and defects sites (Supplementary Figs. 4–7 and Table 1). $^{13}$C NMR spectra validate the loss of original aliphatic $CH_x$ groups (x = 1–3) and functional groups (O–H/N–H) (Fig. 2a). A rather graphitic carbon phase formed, although the signal intensity of aromatic rings reduced due to the lack of hydrogen at the sites[33], consistent with Fourier transform infrared (FTIR) spectroscopy (Supplementary Fig. 8). Raman spectroscopy showed higher intensity ratio of $I_D$ ($A_{1g}$-symmetry, disorder-induced defects) to $I_G$ ($E_{2g}$-symmetry, graphene layer edges) from PDA (0.87) to PDA-C700 (1.01), indicating the increased defects in carbon matrix (Supplementary Fig. 9)[34]. When the temperature was above 700°C, heat energy was provided to grow the graphene lattice, favoring a lower $I_D$ to $I_G$ ratio.

To unveil the change of carbonaceous structure of PDA and PDA-Cx, Powder X-ray diffraction (PXRD) was performed (Fig. 2b). The broad peak of the (002) plane of the graphite phase right shifted from 21.0° of PDA to 22.6° of PDA-C900, revealing the decreased average inter-plane distance (d-spacing) from 4.2 to 3.9 Å based on Bragg's equation[35,36]. The peak of the (100) line at 44.0° does not change position but becomes stronger, indicative of a modest increase in aromatic crystalline size. Positron annihilation lifetime spectroscopy (PALS) is a useful microprobe capable of direct determining the local void spaces and average pore size in a 3D network at the atomic scale[37]. The detailed positron lifetime components ($\tau_1$, $\tau_2$, $\tau_3$) and their corresponding intensities were listed in the supplementary Table 2. The second-component positron lifetime ($\tau_2$) was used to estimate the average pore size (Eq. 1), as nearly no o-Ps component ($\tau_3$) was detected in PALS due to the electronic conductivity of PDA-Cx[38]. PALS-detected average pore size increased from 4.68 Å of PDA-C300 to 5.25 Å of PDA-C900, revealing the formation of slightly larger angstrom-sized voids in carbon matrix induced by higher pyrolysis temperature (Fig. 2c).

**Pore structure analysis via probe gases**
The above PALS technique reflects the variation of average pore size of the accessible free-volume in porous frameworks[39], whereas the probe gases can estimate the size of pore orifices accurately[15]. To further unveil the size distribution of pore orifices, $N_2$ physisorption at 77 K was performed. Negligible $N_2$ uptake (<0.1 mmol g$^{-1}$) was detected in PDA-Cx except for PDA-C700 (Supplementary Fig. 10). However, by further applying probe gases of $H_2$ and $CO_2$, PDA-Cx exhibited remarkably higher $H_2$ and $CO_2$ uptakes at higher pyrolysis temperature, where PDA-C900 realized 5.28 mmol g$^{-1}$ for $H_2$ capture at 77 K and 1.0 bar, and 5.60 mmol g$^{-1}$ for $CO_2$ capture at 195 K and 1.0 bar (Supplementary Figs. 11–12). Above phenomena suggest that smaller sub-5 Å micropores are present in PDA-Cx as the orifice[40,41], to limit $N_2$ to diffuse at critical temperature to the larger voids (4.68-5.25 Å) detected from PALS spectra, but still accessible for the smaller $H_2$ and $CO_2$ probes, which is also supported by wide-angle X-ray scattering (Supplementary Fig. 13). Such a pore system confers PDA-C900 a high surface area up to 400 m$^2$ g$^{-1}$ analyzed by $CO_2$ sorption at 195 K (Supplementary Fig. 14). Nonetheless, the existing theoretical models of PSD for $CO_2$ are relatively insensitive to fit the complex domain structure of carbon[42], as shown by the similar PSD results (Supplementary Fig. 12). Thus, the size distribution of small micropores below 5 Å requires further comprehensive evaluation. Additionally, it was noted that the $CO_2$ adsorption of PDA-Cx at 273 K exhibited less capacity increase at low pressure compared with that at 195 K (Supplementary Figs. 12 and 15), revealing the weak host-guest interaction between $CO_2$ molecules and non-polar carbon surfaces by Van de Waals forces. Inelastic neutron scattering (INS) in the later part can further confirm the weak physisorption. The adsorption is thus reduced to steric restrictions to probe the pore size by various gas molecules with the sub-angstrom size discrepancy as a simplified interpretation.

Specifically, accurate quantification of small micropore orifices in PDA-Cx was resolved by isotherm measurements at 273 K using a series of probing gases ($CO_2$, Ar, $C_2H_4$, $C_3H_6$, $C_2H_6$, $C_3H_8$, $CF_4$, and i-$C_4H_{10}$) with molecular dimensions increasing from 3.3 to 5.0 Å (Fig. 3, Supplementary Figs. 16–20 and Tables 3–6)[43,44]. The pore volume obtained from different probes was estimated by Dubinin-Astakhov (D-A) equation based on the pore-filling adsorption mechanism[45]. A function of the logarithm of adsorbed capacity against the square of potential energy was plotted (Eqs. 2 and 3, Supplementary Figs. 17–22), and the micropore volume can be directly calculated from the intercept (Eq. 4)[46]. Based on the $CO_2$ probe (3.3 Å), the micropore volume of PDA-Cx increased gradually with increased pyrolysis temperature, which reached up to 0.21 cm$^3$ g$^{-1}$ on PDA-C900 (Fig. 3c). Moreover, the difference in micropore volume can be reduced to sterical restrictions in pores smaller than the size of probes, resulting in a high-resolution PSD below 5 Å (Eq. 5). Apart from PDA-C300 with low porosity, the small micropore orifice size of PDA-C500 was determined to be falling in-between 4.1 and 4.7 Å. Higher temperature contributes to the evolution of micropores towards smaller size driven by thermal energy.

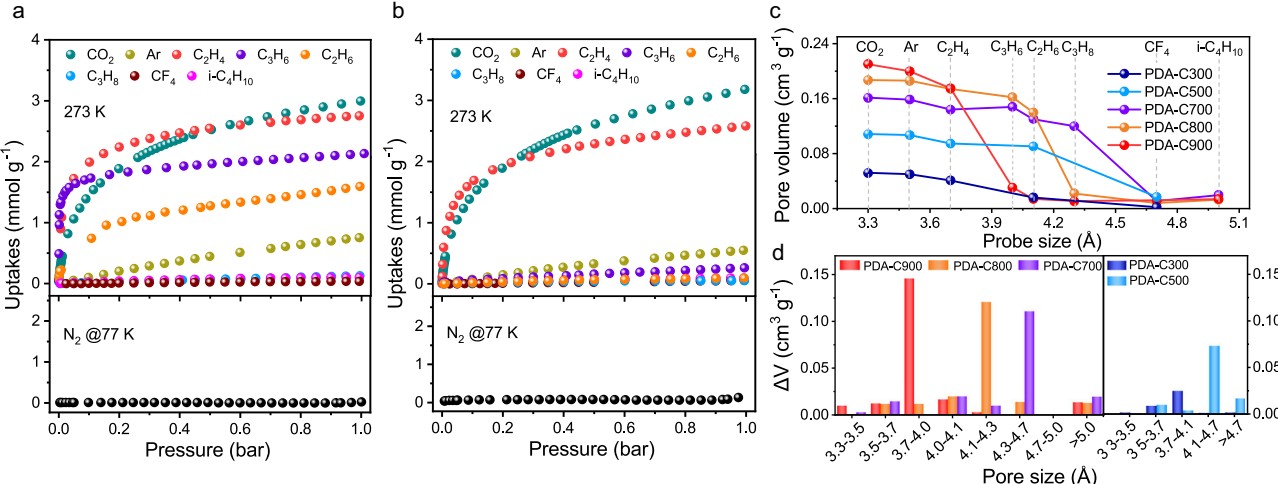

**Fig. 3 | Sorption behavior of gas probes and porosity information.** Sorption equilibrium isotherms of various gas probes with minimum molecular dimensions ranging from 3.3 Å ($CO_2$) to 5.0 Å (i-$C_4H_{10}$) at 273 K and $N_2$ at 77 K on **a** PDA-C800 and **b** PDA-C900 materials; **c** The pore volumes of PDA-Cx calculated from different probe gases based on the Dubinin-Astakhov (D-A) equation (the data points from left to right were calculated from probing gases of $CO_2$, Ar, $C_2H_4$, $C_3H_6$, $C_2H_6$, $C_3H_8$, $CF_4$, and i-$C_4H_{10}$ for PDA-Cx. While $C_3H_6$, $C_3H_8$, and i-$C_4H_{10}$ with high polarizability were excluded for heteroatom-rich PDA-C300 and PDA-C500 due to stronger host-guest interaction); **d** The pore size distributions of PDA-Cx (the differential pore volume (ΔV) of y-axis was obtained from probes with successive sizes. $V_{Ar}$ calculated by Ar was subtracted from $V_{CO2}$, $V_{C2H4}$ from $V_{Ar}$, and so on).

Noticeably, small pore sizes of PDA-C800 and PDA-C900 fell in-between 4.1–4.3 Å and 3.7–4.0 Å, respectively (Fig. 3d). Such narrow size distributions are even comparable with typical crystalline adsorbents such as zeolites and MOFs[3,4,47,48], promising great potential for size-sieving separation of olefin/paraffin. In distinct contrast to the commercial carbons, the major porosities of AC-1 were broadly distributed in the micro-mesopore region ranging from 5.8 to 31.4 Å, while that of CMS-1 were more narrowly distributed in the micropore region of 5.8–15.4 Å, according to $N_2$ physisorption at 77 K (Supplementary Fig. 23). Meanwhile, bare small micropores below 5 Å were probed in CMS-1. A small portion of small micropores but below 3.7 Å was probed in AC-1 (Supplementary Fig. 24). The above encouraging results reveal that the pre-organized precursor and thermal-controlled route could introduce narrowly distributed small micropores below 5 Å in carbons, which was conventionally underestimated due to their non-crystalline nature and absence of molecular-level comprehensive assessment techniques. Furthermore, from the mercury porosimetry and small-angle X-ray scattering, a very small amount of macropores (~1 μm) in PDA-Cx are detected, which can be related to the inter-particle voids that barely contributes to the adsorption of angstrom-size gases (Supplementary Figs. 25–26).

### Molecular recognition of olefin/paraffin

To evaluate the adsorptive separation performance of PDA-Cx, single-component gas ($C_2H_4$, $C_2H_6$, $C_3H_6$, and $C_3H_8$) sorption isotherms were collected and satisfactorily described by model-fitted approach (Supplementary Figs. 27–37). Non-porous PDA and low-temperature pyrolyzed PDA-C300 exhibited negligible capacity of both $C_2$- and $C_3$-olefin/paraffin pairs (<0.2 mmol $g^{-1}$ at 1.0 bar and 298 K) (Supplementary Figs. 27–29). The pyrolysis of PDA at higher temperatures is responsible for developing accessible micropores for adsorption. Specifically, PDA-C500 and PDA-C700 showed co-adsorption of both $C_2$ and $C_3$ components due to the enlarged size of the micropore orifice (Supplementary Fig. 30–33). As pyrolysis temperature increased further, more refined small micropores started to form. Narrowly distributed pore orifices of 4.1–4.3 Å in PDA-C800 fell in-between the minimal molecular dimension of $C_3H_6$ and $C_3H_8$, allowing the entry of $C_3H_6$ with a capacity of 1.98 mmol $g^{-1}$ at 1.0 bar and 298 K, and exclusion of larger $C_3H_8$ molecules with separation factor (S') reaching up to 36.7 (Fig. 4a). On the other hand, the S' value of $C_2H_4$/

$C_2H_6$ for PDA-C900 with narrower micropore orifice (3.7–4.0 Å) could reach 24.7, with a high $C_2H_4$ uptake of 2.25 mmol $g^{-1}$ at 1.0 bar and 298 K, and negligible $C_2H_6$ uptake (Fig. 4b). The superior selectivity outperforms previously reported carbonaceous materials (Fig. 4c), as well as most zeolites and MOFs (Supplementary Fig. 38 and Tables 7–8). Conversely, AC-1 and CMS-1 with widely distributed pores exhibited a similar capacity for both $C_2H_4$/$C_2H_6$ and $C_3H_6$/$C_3H_8$, and a low S' value of <1.1 (Supplementary Figs. 39–40). The mainly reported approach to enhance S' value to 3.8 for $C_2H_4$/$C_2H_6$ separation and 3.5 for $C_3H_6$/$C_3H_8$ separation is by introducing the pi-complexation function of monovalent copper or silver. However, such an approach is hindered by higher heat of adsorption requiring more energy for desorption and the undesired polymerization of adsorbent in long-runs[49,50]. The above results suggest that the tailorable narrow micropores in carbon are highly desirable for the selective separation of $C_2$/$C_3$ olefin/paraffin pairs.

Inelastic neutron scattering (INS) measurements were performed to probe the interaction between $C_3H_6$ and $C_2H_4$ and the pore surface in the PDA-C800 and PDA-C900 substrate (Fig. 4d, e)[51]. By comparing the vibrational spectrum of solid $C_3H_6$ with the spectrum of PDA-C800 with $C_3H_6$ after the PDA-C800 spectrum is subtracted, we can see that below 140 $cm^{-1}$, discrete phonon modes are apparent in the solid $C_3H_6$ spectrum (Fig. 4d). However, these modes are less prominent in the adsorbed $C_3H_6$ sample, indicating no solid $C_3H_6$ formed inside the PDA-C800 sample. In addition, three prominent vibrational modes are observed in the vibrational spectrum around 225, 425, and 580 $cm^{-1}$. Only the mode at around 580 $cm^{-1}$ is shifted compared to solid $C_3H_6$, and this mode is also slightly broadened. This would indicate weak physisorption of $C_3H_6$ on the porous carbon framework and is likely related to the torsion of the methyl group of the guest molecules and deformation of the =$CH_2$ group. Perturbation of the mode (shift and broadening) is consistent with the $C_3H_6$ molecule lying flat on the carbon surface. The modest shift would indicate a weak interaction with the carbon surface hence supporting that the selectivity in PDA-C800 is primarily a size effect rather than active, strong physisorption. The split modes around 225 $cm^{-1}$ are the methyl torsions and are consistent with the solid $C_3H_6$, where the splitting is due to inter-molecular interactions. Given the weak interaction of $C_3H_6$ with the carbon framework, it is likely that the splitting is due to intermolecular interactions of neighboring $C_3H_6$ molecules in the pores. Similar

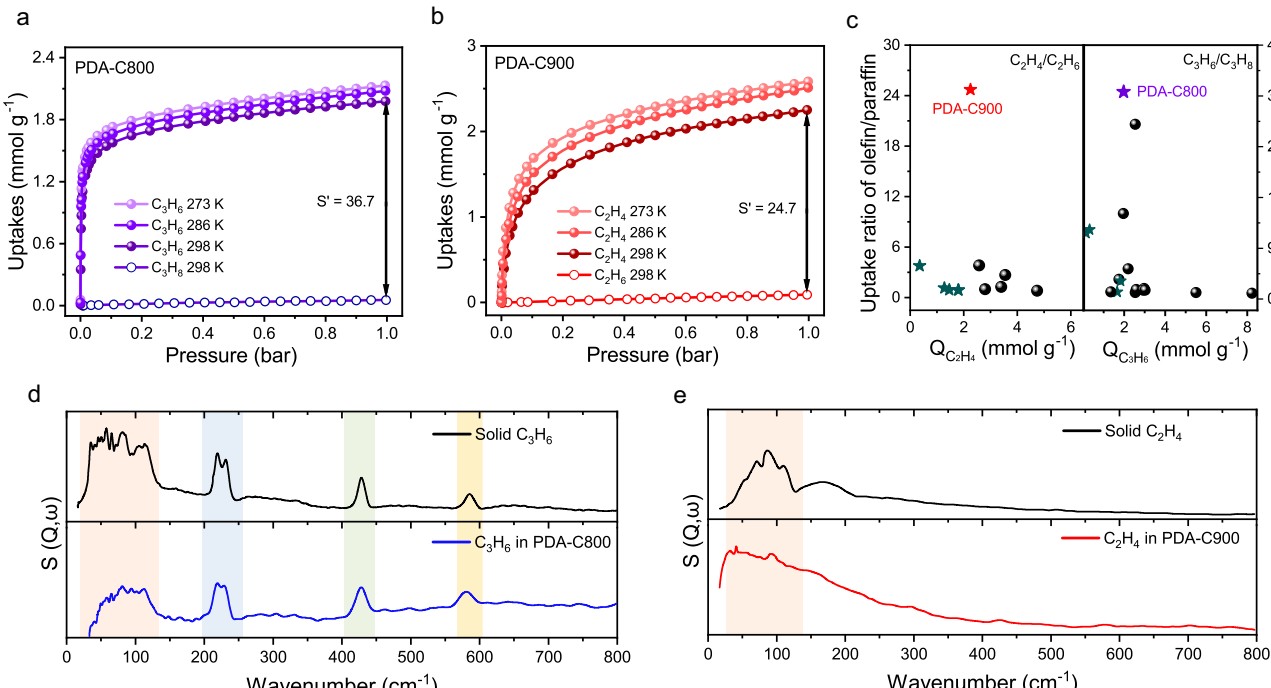

**Fig. 4 | Static sorption property of olefins and paraffins.** Single-component gas adsorption isotherms of **a** $C_3H_6$ and $C_3H_8$ on PDA-C800 and **b** $C_2H_4$ and $C_2H_6$ on PDA-C900 at varied temperatures. The separation factor (S') was calculated by the uptake ratio of olefin/paraffin at 298 K and 1.0 bar; **c** Comparison of the number of olefins adsorbed (Q) and olefin/paraffin separation factor of PDA-Cx with state-of-the-art porous carbon adsorbents (PDA-Cx marked as pentagram). The details are given in Supplementary Tables 7-8. **d** Comparison of the experimental INS of solid $C_3H_6$ and $C_3H_6$ in PDA-C800 (difference between PDA-C800 and PDA-C800 with $C_3H_6$ adsorbed); **e** Comparison of the experimental INS of solid $C_2H_4$ and $C_2H_4$ in PDA-C900 (difference between PDA-C900 and PDA-C900 with $C_2H_4$ adsorbed). Discrete phonon modes are highlighted by orange color, and three prominent vibrational modes are highlighted by blue, green and yellow colors.

observations can be made for $C_2H_4$ adsorbed in PDA-C900, where the absence of prominent discrete phonon modes indicates weak interaction between $C_2H_4$ molecules and the carbon substrate (Fig. 4e). The broad phonon band indicates no bulk, solid $C_2H_4$ formed. However, the simplicity of the vibrational spectrum of $C_2H_4$ limits further observations.

## Dynamic separation of olefin/paraffin

Besides static adsorption, single-component diffusional kinetics is another important parameter related to practical application[52,53]. PDA-C900 reached 80% of the saturated $C_2H_4$ uptake and equilibrium within 27 and 85 min, respectively, with the calculated diffusion rate (D' = D $r^{-2}$) of $4.13 \times 10^{-3}$ min$^{-1}$ (Fig. 5a, Supplementary Fig. 41). Bare $C_2H_6$ adsorption (<0.01 mmol g$^{-1}$) was found within the whole 100 min. Such molecular-recognition behavior is consistent with static isotherms. As expected, PDA-C800 reached 80% of the $C_3H_6$ saturated uptake and equilibrium within about 12 and 36 min, respectively, with a diffusion rate of $7.57 \times 10^{-3}$ min$^{-1}$. The $C_3H_8$ uptake is negligible in the process (<0.01 mmol g$^{-1}$) (Fig. 5a). The moderate adsorption rate of olefins is beneficial for actual dynamic olefin/paraffin separation.

Breakthrough experiments were conducted to validate the feasibility of separating binary mixtures under dynamic conditions (Supplementary Figs. 42–47, Tables 9–10). The molecular-recognition carbons, including PDA-C800 and PDA-C900, were chosen as targeted materials for the systematic study. A clean and sharp sieving separation of the $C_3H_6$/$C_3H_8$ mixture was realized over PDA-C800 with a $C_3H_6$ breakthrough time of around 26 min (Fig. 5b). The calculated dynamic $C_3H_6$ uptake was 1.5 mmol g$^{-1}$. As expected, a clean separation of the challenging $C_2H_4$/$C_2H_6$ mixture was observed by the fixed bed packed with PDA-C900. $C_2H_6$ eluted out at the beginning owing to complete exclusion, and smaller $C_2H_4$ was retained until around 30 min (Fig. 5c). The calculated dynamic uptakes of $C_2H_4$ reached 1.8 mmol g$^{-1}$; Subsequent desorption of eluted gas by heating under helium flow yielded

$C_2H_4$ and $C_3H_6$ purity larger than 97.2% and 98.1%, respectively, before the period of 60 min (Fig. 5d). In contrast, though AC-1 and CMS-1 provided a longer $C_2H_4$ or $C_3H_6$ retaining time, the similar breakthrough time led to a lower separation efficiency (Supplementary Figs. 43–46). Above results further highlight the importance of refined small micropores for highly efficient molecular recognition of gas mixture. Multiple adsorption-desorption cycles demonstrated the excellent stability of PDA-Cx (Fig. 5e). The easier renewability can be quantitatively demonstrated by the low isosteric heat of adsorption ($Q_{st}$) of 32.3 and 41.5 kJ mol$^{-1}$ for $C_2H_4$ in PDA-C900 and $C_3H_6$ in PDA-C800 at zero loading, respectively (Fig. 5f, and Supplementary Table 11).

## Discussion

In conclusion, precise probing and manipulating narrowed small micropores in carbons unveiled the existing blind spot of pores below 5 Å and verified their efficiency in important industrial gas separations. PDA-Cx possesses refined sub-5 Å small micropore orifices together with 4.68–5.25 Å microvoids from thermal-controlled pyrolysis of polydopamine. Though undetectable by the standard $N_2$ adsorption at 77 K, the small micropores below 5 Å were accurately probed and defined by a series of gas molecules ranging from 3.3 to 5.0 Å at sub-angstrom size discrepancies. Results show that the small micropore orifices of 3.7–4.0 Å in PDA-C900 and 4.1–4.3 Å in PDA-C800 fall in-between the molecule size of $C_2H_4$/$C_2H_6$ and $C_3H_6$/$C_3H_8$ pairs, respectively, allowing the entrance of olefins while entirely exclude larger paraffin counterparts. The larger voids allow high $C_2H_4$ and $C_3H_6$ uptakes of 2.25 and 1.98 mmol g$^{-1}$ at 298 K and 1.0 bar, respectively. Binary breakthrough experiments have further demonstrated ideal molecular recognition behavior. We identified the weak physisorption of olefins in carbon through INS measurement, supporting the size-sieving effect rather than strong enthalpic interaction. This work provides important indications for designing and probing carbon-

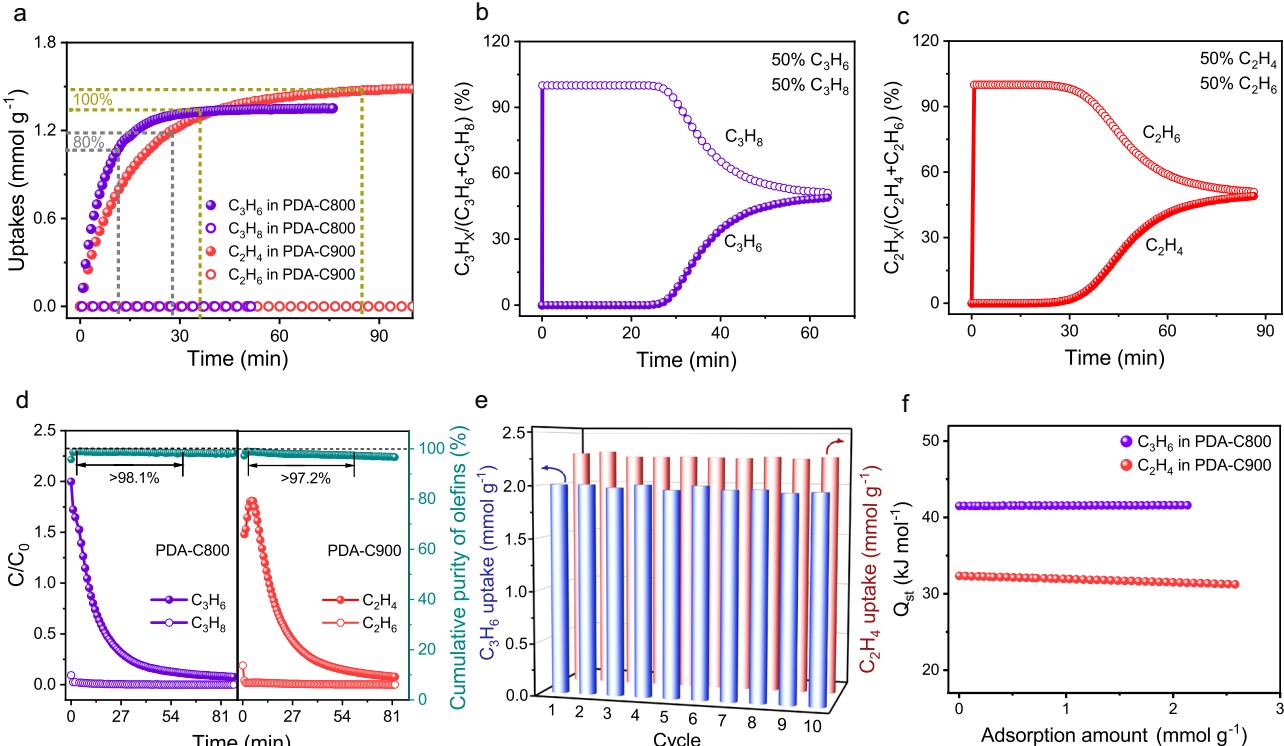

**Fig. 5 | Olefin/paraffin dynamic separation and regeneration property. a** Time-dependent gas uptake profiles of $C_3H_6$ on PDA-C800 and $C_2H_4$ on PDA-C900 at 308 K and 0.5 bar. The time required to reach 80% of the saturated olefin uptake and equilibrium is depicted as dashed lines. Experimental column breakthrough curves for the mixtures of **b** $C_3H_6/C_3H_8$ (50/50, v/v) on PDA-C800 and **c** $C_2H_4/C_2H_6$ (50/50, v/v) on PDA-C900 at 298 K with a constant flow rate of 1.5 mL/min; **d** Desorption curves following a column breakthrough experiment under 10 mL/min flow of He at 353 K with PDA-C800 and PDA-C900, and the corresponding cumulative purity of olefins. C and $C_0$ are the concentrations of each gas at the outlet and inlet, respectively; **e** Cycling uptake tests for pure-component $C_3H_6$ on PDA-C800 and $C_2H_4$ on PDA-C900; **f** Heat of adsorption ($Q_{st}$) of $C_3H_6$ in PDA-C800 and $C_2H_4$ in PDA-C900.

based materials with finely tunable small micropores below 5 Å that can also be extended to other challenging separations, as well as catalysis, energy storage, and conversion.

# Methods

## Synthesis of PDA-Cx

In a typical synthesis, 1.5 ml $NH_4OH$, 180 ml deionized water (DI) and 80 ml $C_2H_5OH$ were mixed thoroughly to form solution A. Then, 2 g dopamine hydrochloride was dissolved in 20 ml DI water as solution B. Next, solution B was added dropwise to solution A under continuous stirring and reacted for 30 h at room temperature. Next, the black polydopamine product (named PDA) was collected by vacuum filtration, followed by washing with DI water three times and drying at 60 °C overnight. Afterward, the PDA product was placed in a tube furnace, and carbonized at different temperatures (300–900 °C) for 1 h under nitrogen atmosphere with a heating rate of 5 °C min⁻¹. The resulting products were PDA-Cx (where x stands for the final carbonization temperature).

## Positron annihilation lifetime spectroscopy (PALS) experiment

PALS was performed to analyze the free-volume of the samples. The positron source (²²Na) with an activity of 30 μCi was used and sandwiched between two same disk-shaped pieces. The sample-source-sample set was kept at room temperature in a vacuum chamber (vacuum better than $1 \times 10^{-3}$ Pa) during the PALS measurements. One gamma detector is regarded as "start" to detect the 1.275 MeV gamma from the ²²Na nucleus that is emitted simultaneously with the beta decay positron. Another gamma detector is "stop" for detecting one of the two subsequent 0.511 MeV annihilation photons. Electronics measure the "start" - "stop" time intervals between two γ rays, and then lots

of annihilation events were recorded for a positron lifetime spectrum for eventual computer fitting. Each lifetime spectrum contains about $5 \times 10^6$ accumulated counts. The time resolution of the digital PALS spectrometer (TechnoAP, Japan) was 190 picosecond (ps) and the count rate was around 200 cps. The PALS spectra were analyzed using the LTv9 program.

In our work, the o-Ps intensity ($I_3$, formation probability of o-Ps) in each PDA-Cx sample is lower than 1% (Supplementary Table 2). Thus, the average diameter (2 R) of free-volumes was estimated from $\tau_2$ based on an empirically linear equation, that exhibits a decent correlation coefficient of 0.9268 for $\tau_2$ -R data on polymers, zeolites, and molecular sieves with R < 5 Å[38]. The equation is formally written as:

$$\tau_2 = 0.174(1 + 0.494R) \qquad (1)$$

Where, the units of $\tau_2$ and R are expressed in ns and Å, respectively.

## Pore size distribution calculation

This part used various sorbates ($CO_2$, Ar, $C_2H_4$, $C_2H_6$, $C_3H_6$, $C_3H_8$, $CF_4$, i-$C_4H_{10}$) with different molecular dimensions probe molecules to determine the pore size distribution and pore volume[54,55]. The physical property of molecular probes was listed in Supplementary Table 3. Their isotherms were measured at 273 K volumetrically by using a Micromeritics 3Flex. The adsorption data were correlated by the Dubinin-Astakhov (D-A) equation, which modified the Dubinin-Radushkevich (D-R) equation. The D-R equation predicted adsorption isotherms on homogeneous pore surfaces, while the D-A equation evaluated the adsorption isotherms on micropore surfaces with some

degree of heterogeneity[56]. D-A equation is formally written as[57]:

$$W = W_0 \exp\left[-\left(\frac{A}{\beta E_0}\right)^n\right] \qquad (2)$$

Here, W refers to the adsorbed capacity (mmol g$^{-1}$), $W_0$ is the saturation capacity (mmol g$^{-1}$), $\beta$ is termed the similarity affinity coefficient of the characteristic curves, $E_0$ is the micropore characteristic energy of adsorption, and n is an empirical constant related to surface heterogeneity. If $n = 2$, the D-A equation reduces to the D-R form. A is the adsorption potential by Polanyi, which can be defined as:

$$A = RT \ln\left(\frac{P_0}{P}\right) \qquad (3)$$

$P_0$ and P represent saturated vapor pressure and adsorption pressure (bar), respectively. T is the temperature (K), and R is the gas constant. Therefore, $W_0$ can be calculated from the intercept of the line plot of ln W vs ln$^n$ ($P_0$ P$^{-1}$). Furthermore, $W_0$ is defined as:

$$W_0 = V_0 \rho_M \qquad (4)$$

$V_0$ is the micropore volume (cm$^3$ g$^{-1}$) and $\rho_M$ is the liquid molar density (mmol cm$^{-3}$). Thus, by combing Eqs. (2–4), a series of micropore volumes ($V_0$) calculated from different gas probes can be obtained. After that, the $V_0$ calculated from gas A was subtracted from $V_0$ calculated from gas B. In this way, the differential pore volumes can be obtained, as shown below:

$$\triangle V = V_{0(\text{gas A})} - V_{0(\text{gas B})} \qquad (5)$$

$\Delta$V is the pore volume with a size falling in-between the molecular size of gas A and gas B.

## Measurement of statistic adsorption isotherms

The system is connected to a vacuum station that can generate a vacuum up to $10^{-7}$ Pa. $N_2$ and $H_2$ physisorptions at 77 K were performed on a Micromeritics ASAP 2020 with liquid nitrogen as a coolant. $CO_2$ physisorption at 273 K and 195 K were performed on a Micromeritics ASAP 2020 with ice water and dry ice/isopropanol as coolants, respectively. Olefins and paraffin physisorption at different temperatures (273, 286, and 298 K) were determined using a Micromeritics 3Flex and ice water or a recycle water bath was used to control the temperature. Before the measurements, about 100 mg of the samples were degassed under a vacuum at 150 °C for at least 6 h.

## Measurement of adsorption kinetic curves

Time-dependent adsorption measurements of $C_3H_6$, $C_3H_8$, $C_2H_4$, and $C_2H_6$ were performed using a TGA 55 thermogravimetric analyzer (TA Instruments, USA). Before the measurement, the sample was degassed at 423 K for 6 h under the $N_2$ flow. After that, single-component gas of $C_2H_4$, $C_2H_6$, $C_3H_6$, or $C_3H_8$ flowed through the electrobalance system at 0.5 bar. The analyzer temperature was controlled at 308 K. Adsorption kinetic curves were done until the adsorption capacity reached saturation. Diffusional time constants (D', D r$^{-2}$) were calculated by the short-time solution of the diffusion equation[58]:

$$\frac{q_t}{q_\infty} = \frac{6}{\sqrt{\pi}} \times \sqrt{\frac{D}{r^2} \times t} \qquad (6)$$

Where $q_t$ is the adsorbed capacity at time t, $q_\infty$ is the adsorbed capacity at equilibrium, D is the diffusivity and r is the radius of the equivalent spherical particle. The slopes of $q_t/q_\infty$ versus $t^{1/2}$ are derived from the

fitting of the plots in the low gas uptake range, and then D' can be calculated from the square of the slope multiplied by $\pi/36$.

## Double-site Langmuir isotherms model

The adsorption isotherms data of olefins and paraffin with high capacity (>0.15 mmol g$^{-1}$ at 1.0 bar) were fitted with the double-site Langmuir isotherms model. The model is shown as follows:

$$Q = q_{\text{A,sat}} \frac{b_A p}{1 + b_A p} + q_{\text{B,sat}} \frac{b_B p}{1 + b_B p} \qquad (7)$$

Where, Q is the adsorbed capacity (mmol g$^{-1}$), $b_A$ and $b_B$ refer to the corresponding adsorption equilibrium constants reflecting the affinity coefficients of sites A and B, Pa$^{-v}$, respectively. $q_{\text{A,sat}}$ and $q_{\text{B,sat}}$ are the saturation adsorbed capacity of sites A and B; P corresponds to an equilibrium pressure (bar). The fits are of good accuracy ($R^2 > 0.971$).

## Langmuir-Freundlich isotherms model

The adsorption isotherms data of olefins and paraffin with very low capacity (<0.15 mmol g$^{-1}$ at 1.0 bar) were fitted with the Langmuir-Freundlich isotherms model. The model is shown as follows:

$$Q = q_{\text{A,sat}} \frac{b_A p^n}{1 + b_A p^n} \qquad (8)$$

Where, Q is the adsorbed capacity (mmol g$^{-1}$), $b_A$ refers to the corresponding adsorption equilibrium constants reflecting the affinity coefficients, Pa$^{-v}$. $q_{\text{A,sat}}$ is the saturation adsorbed capacity; P corresponds to an equilibrium pressure (bar); n represents the deviation from the ideal homogeneous surface. The fits are of good accuracy ($R^2 > 0.991$).

## Inelastic neutron scattering (INS) measurements

INS data was collected on the VISION Beamline at Oak Ridge National Laboratory. Firstly, 1 g of each sample was activated by heating to 423 K for 2 h under a vacuum. Next, each sample was measured empty (absent of guest molecules) and subsequently dosed with 1 mmol ($C_3H_6$ for PDA-C800 and $C_2H_4$ for PDA-C900) at room temperature. After equilibrating the samples for 1 h, INS data were obtained for the samples in the presence of the guest molecules. Data collection for the empty and dosed samples took place at 5 K to reduce thermal effects. Finally, the data from the empty materials was subtracted from that obtained from the dosed samples to remove the porous carbon contribution and investigate the signal from the adsorbed gases.

## Breakthrough experiments

The breakthrough experiments were performed by using a laboratory-built setup (Supplementary Fig. 47). The composition of the feed gas stream is as follows: $C_3H_6/C_3H_8$ (50/50, v/v) and $C_2H_4/C_2H_6$ (50/50, v/v). 950 mg of activated samples were packed into a stainless-steel column (4.6 mm inner diameter), and the gas flow rate was controlled at 1.5 ml min$^{-1}$ by mass flow meter. The gas stream at the outlet gas was monitored by using gas chromatography (GC-9560) with a flame ionization detector (FID). Desorption data were collected under 10 mL min$^{-1}$ flow of He at 353 K with adsorb saturated samples.

## Reporting summary

Further information on research design is available in the Nature Portfolio Reporting Summary linked to this article.

# Data availability

The authors declare that all data supporting the conclusions of this work are available in the article and its Supplementary Information. Further source data will be made available at request from the corresponding authors.

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

## Acknowledgements

Funding: J.X., S.-J.D., J.H., and C.Y. gratefully acknowledge the financial support from the National Natural Science Foundation of China (22022806). M.R.R. and S.D. acknowledge the U.S. Department of Energy (DOE) Office of Science, Office of Basic Energy Sciences, Chemical Sciences, Geosciences, and Biosciences Division (Separation Sciences) for research funding to support the neutron scattering experiments and data analysis. This research used resources at the Spallation Neutron Source (SNS), a DOE Office of Science User Facility operated by Oak Ridge National Laboratory.

**Note** This manuscript has been authored by UT-Battelle, LLC, under contract DE-AC05-00OR22725 with the US Department of Energy (DOE). The US Government retains and the publisher, by accepting the article for publication, acknowledges that the US government retains a non-exclusive, paid-up, irrevocable, worldwide license to publish or reproduce the published form of this manuscript, or allow others to do so, for US government purposes. DOE will provide public access to these results of federally sponsored research in accordance with the DOE Public Access Plan (http://energy.gov/downloads/doe-public-access-plan).

## Author contributions

J.X. conceived the project. J.X., S.D., and B.C. co-supervised the project. S.-J. D. and J.H. designed the experiments. S.-J. D. conducted the breakthrough experiments, analyzed the data, and wrote the manuscript. J.H. prepared the samples and performed the initial experiments. M.R.R. and L.L.D. carried out the INS experiments and analyzed the results. H.Z. conducted the PALS experiments. P.Y. and Y.L. conducted the SAXS and WAXS experiments. C.Y. helped in drawing figures and adsorption data analyses. All authors discussed the results and commented on the manuscript.

## Competing interests

The authors declare no competing interests.
