## [Peer Review File · Nature Communications]

Probing sub-5 Ångstrom micropores in carbon for precise light olefins/paraffins separationReviewer comments, first round review –

Reviewer #1 (Remarks to the Author):

The authors in this manuscript discovered that the carbon prepared by parolysis of polydopamine contained hidden narrow ultramicrochannels that cannot be discovered by normal nitrogen adsorption isotherm, and the obtained carbon showed excellent adsorption separations for ethane/ethylene and propane/propylene. The authors further revealed that the separation is mainly based on size effect, not due to molecular interaction. The result is somewhat unexpected because carbon materials prepared by parolysis do not expect to give narrow pore size distribution and thus good separation performance. The reported olefin/paraffin separations are very important in industry. Most of the authors are experts in the field. The work is solid. I recommend to accept the manuscript as it is for timely publication.

Reviewer #2 (Remarks to the Author):

The manuscript "Probing "Hidden" Narrow Ultramicrochannels in Carbon for Precise Light Olefins/paraffins Separation" is an interesting and valuable study of the materials promising to be efficient sieves for the separation of alkanes and alkenes. The breakthrough tests show that the selectivity, uptakes and other important for separation properties are comparable to, e.g. functionalized metal organic frameworks. However, relatively simple synthesis and stability of the polydopamine-based microporous carbons make them a considerable competition for other materials. The breakthrough tests are preceded by a careful characterization of the studied materials. Multiple techniques give a quite comprehensive knowledge of their structure and porosity. Especially, the porosity characterization is a demanding task for the pores of such a small size. Taking these advantages into account I would recommend the manuscript for publication, but the authors did not avoid a few inaccuracies or mistakes that should be corrected. I should point out that further criticism of the terminology concerns the scientific community rather than authors who have had to succumb to common trends. Beginning with the title, the requirement of self-advertising and drawing attention negatively affects the clarity and precision of the language. The word "hidden" does not seem proper to describe the pores. The inability to detect them by the LN₂ sorption does not make them hidden. The authors prove by themselves that the pores are not hidden, but just require more sophisticated approach to characterize them.

Another inaccuracy is using "ultramicrochannels" to describe the pores. First of all, there is no premises presented to determine that the pores are channels (i.e. they are roughly cylindrically shaped). Moreover, "ultra" is not a strict definition. According to the most popular IUPAC classification micropores have the aperture smaller than 2.0 nm (often defined in the range of 0.3 nm to 2.0 nm). But "ultramicro" (even though a few times used in the scientific literature) has no clear meaning. This resembles exaggerating suffixes in advertisements (like "hyper" or "super"). Finally, I take this opportunity to provide a comment for general consideration (not necessarily changes to the text). Topics, methods, materials etc. are often referred to as "new" or "novel" in modern scientific papers. This is not found in older papers, while their contribution to the progress of science is unquestionable. Do we have to emphasize importance of our works so directly? Except these few remarks of general nature I would like to draw attention to the more significant inaccuracies. Interpreting the PALS results is unlike any approach to analyzing such data that I have seen before. Although Zgardzińska recently promotes presenting data in similar way (INTI plots), fig.2d definitely does not present lifetimes and intensities of free positrons, because there should be a single intensity and lifetime for each sample. The positron lifetime is a parameter that describes the rate of an exponential decay (like in radioactive decays). There are usually several (rarely one) components in a PAL spectrum, so describing the x-axis in Fig.2d as "Lifetime" is hardly correct if the plot consists of dozens of points. In turn, FWHM is reserved for the spectrometer's time resolution function and cannot be dimensionless, so the inset also needs to be revised. In fact, Fig.2d resembles PAL spectra plotted in an uncommon lin-log scale (x-axis should be described as "Time" in such case). In other papers spectra are usually shown in log-lin scale, but most often not shown at all due to their visual similarity. Only numerical

analysis with the use of dedicated software (PALSfit, LT, MELT) allows to determine component intensities I_i , lifetimes τ_i and their distributions o_i (i enumerates components). The results are discussed on this basis. In "Methods. Positron annihilation lifetime spectroscopy experiment." the authors describe using LT9, but I do not see any results of such analysis. The "Methods. Positron annihilation lifetime spectroscopy experiment." part also require some improvement:

- Description "electronics" is not sufficient for PAL spectrometer.

- I cannot agree with the statement: "the value of τ_{3} is low due to the weak electronic conductivity of the pyrolyzed samples". Positronium (usually the third component) is not formed (or quenched with high rate) in conductive samples, not the other way around, as the authors write.

- What is (s_3) described as the third component? The symbol is the most similar to σ_{3} , which is the distribution of lifetimes.

So PALS results cannot be accepted in the present form. However, if fig. 2d presents the spectra, even their visual differences allow to expect that interesting and valuable information can be obtained from PALS. Therefore, I encourage the authors to analyze the PALS data properly and draw conclusions from the results.

It is not clear what authors mean by limitation of diffusion in description of LN_{2} sorption. In addition to diffusion, adsorption involves several other phenomena. Here, I would expect that not only quite arbitrary defined molecule sizes, but also the energies of the adsorbate-adsorbent adsorbate-adsorbate interaction play important role. Probably limiting it to van der Waals forces is also a simplification. As the authors correctly write, simple adsorption models do not work for these pore sizes and the example of CO_2 adsorption at different temperatures confirm the importance of interaction energy. Therefore reducing the discussion of adsorption to steric restrictions as Fig.S13 is a simplified interpretation (even though it is convenient and easy to understand). This should be clearly written to avoid misleading a reader. Neutron results shed some light on this, but they are presented many paragraphs later. Probably they should be at least mentioned or briefly announced in discussion of LN_{2} sorption.

The size deviation of the PUC-800 and PUC-900 pores (estimated by Dubinin-Astakhov equation), AC-1 and CMS-1 (from LN_{2} sorption) is not clearly defined. It does not seem to be a standard deviation or FWHM. Moreover, comparing pore sizes and their distributions obtained by different methods for these samples is not proper. Especially keeping in mind previously presented failure to determine pore sizes for PUC with LN_{2} . I wonder why the same adsorption probes as for PUC were not used for AC-1 and CMS-1 – comparison of such results for small pores would be interesting and would allow to verify PUC results.

I could not find any explanation how the pore size distributions presented in Fig.3d and S18b were calculated.

Mercury porosimetry is suitable to find macropores not mesopores. Moreover it raises suspicions if high pressure influences measured values. Especially in the range of smaller pores.

The description of the results in the chapter "Dynamic separation of olefins/paraffins", i.e. 80% and equilibrium times cannot be read from fig. S37 and S38. What "Sqt" in this figure means? Maybe this was meant "sqrt", "square root"? In such case I suggest to present data on the square root scale (or log scale), where time is given in minutes, not in square root of minutes. Slightly increasing value in fig. S37 does not support the sentence "No detectable C_2H_6 uptake was found within the whole 100 min" (and 100 min confirms a sqrt scale here). This is a little confusing. What are uncertainties of the calculated diffusion rates? They can be easily calculated from fitting.

The description of models in "Methods. Molecular probe calculation" does not lead to $\Delta V(D)$, which is presented in figures. I suggest to extend this description to show the full course of calculations. What are n values in fig. S17?

Last part of my comments concern clarity and ease of reading of the manuscript.

Using the Supplementary Materials is not convenient. A reader has to read both documents simultaneously to understand the text. Either some of the discussion referring to figures in SI should be moved to SI and only simple conclusion should be left in main text or the figures from SI should be moved to the main text.

The structural formulas in fig.2a are very small.

Reading comparison of the sorption of olefins/paraffins I miss the figure with comparison of the pore sizes of all PUCs made by the same method (from the Dubinin-Astakhov equation or a little mysterious calculation based on the size of gas probes). Even these presented for PUC-700, PUC-800 and PUC-900 are divided to main text and the Supplementary Materials.

I suggest to enlarge figures in Supplementary Materials where possible – currently high

magnifications of the document are required to see their details.

There is no labels above the break of y-axis in fig. S34, which does not allow to read values on this figure without looking at fig. 4c.

Reviewer #3 (Remarks to the Author):

The authors present a study on the ultramicropores of carbons that are not easy to find or access. Therefore, they prepare materials from pre-ordered polymeric material. Many measurements were performed to investigate the nature of those pores, ranging from gas sorption with species such as CO₂ and O₂ up to bigger molecules, but also synchrotron methods were used such SAXS, WAXS and positron annihilation lifetime spectroscopy (PALS). The found pore-sizes in the range of 3-5 Å seemed to be interesting for olefin paraffin separations, especially focusing on propylene/propane separation. The sorption properties were probed, and weak physisorption for the probe molecules was found using neutron scattering. Therefore, the authors apply a breakthrough analyzer to determine separation performance.

Some questions arise from reading the manuscript:

1) In your introduction you mention the reference 24, which has proven the hidden micropores. You are adapting their method but are not able to reproduce it for your samples correctly. Somehow, I am missing the part where you are mentioning the pore volumina for the micropores in the paper. Table S2 has them, but they are not referred to in the main text – because they are super low. In the reference-paper, those volumes are also extremely low and this is the reason why these pores are “hidden” – you have to look very exactly at the material. If they are that low in your material, there is no way these channels are responsible for the strong separation values you measure in the end. Please re-evaluate your methods especially for PUC-900. (see also comment 2)

2) In your PUCs you find macropores and tried to proof the “hidden” micropores inside your carbons through various methods, which you then conclude to be the factor for high separation vallues. Further, you proof that there is not much adsorption happening on your carbons. Nevertheless, you find good separation values for paraffin/olefin separation in your carbon materials. Now, changes in breakthrough are rather influenced by sorption than size-siving. Thus, if there are lots and lots of larger pores available and no sorption selectivity can be found, why should this material be so good in this type of gas separation? Please explain to me how your data combines rational to the effect you find in breakthrough.

3) Looking at this interference of your own data, where on the one hand, almost no favourable sorption property of your carbons is found and the micropores are so well hidden (or only very low amounts of those pores are in the material, which would be my ration explanation in accordance with ref.24), but on the other hand the superior separation values are measured, I find it hard to acknowledge these properties to size sieving effects.

4) This further makes me think about your conclusion: Might this not be a completely different effect? Why do you compare your material to size-sieving MOFs? The introduction should be rewritten, and the focus should lie on size-sieving carbons rather than other materials. The benchmarking is fine, but if this is not a size-sieving effect, maybe a different benchmark needs to be considered.

5) Maybe, an easy way to proof the amount of micropores to the total volume of your carbons would be Archimedes density measurements. In gas pycnometer, you would be able to measure densities of your materials outgoing from He and with the correct way of calibration you can go up to SF₆. These “skeletal” density could be considered to estimate the volumetric amounts of your micropores.

Response to Reviewers of Manuscript NCOMMS-22-40515-T

The authors would like to thank the Reviewers for their constructive comments and suggestions on our manuscript entitled “Probing Sub-5Å micropores in Carbon for Precise Light Olefins/paraffins Separation” (previously as “Probing “Hidden” Narrow Ultramicrochannels in Carbon for Precise Light Olefins/paraffins Separation”). We have carefully considered all the Reviewers’ comments and have revised the manuscript to address their concerns. To aid in the reviewing process, we have replied to all the comments on a point-by-point basis and marked all changes in the manuscript text file with track changes. We hope the manuscript can now be accepted for publishing in *Nature Communications*.

Reviewer(s)' Comments to Author:

Reviewer: 1

Comments:

The authors in this manuscript discovered that the carbon prepared by pyrolysis of polydopamine contained hidden narrow ultramicrochannels that cannot be discovered by normal nitrogen adsorption isotherm, and the obtained carbon showed excellent adsorption separations for ethane/ethylene and propane/propylene. The authors further revealed that the separation is mainly based on size effect, not due to molecular interaction. The result is somewhat unexpected because carbon materials prepared by pyrolysis do not expect to give narrow pore size distribution and thus good separation performance. The reported olefin/paraffin separations are very important in industry. Most of the authors are experts in the field. The work is solid. I recommend to accept the manuscript as it is for timely publication.

We highly appreciate the reviewer’s positive comments and recommending it for timely publication.

Reviewer: 2

Comments:

The manuscript "Probing "Hidden" Narrow Ultramicrochannels in Carbon for Precise Light Olefins/paraffins Separation" is an interesting and valuable study of the materials promising to be efficient sieves for the separation of alkanes and alkenes. The breakthrough tests show that the selectivity, uptakes and other important for separation properties are comparable to, e.g. functionalized metal organic frameworks. However, relatively simple synthesis and stability of the polydopamine-based microporous carbons make them a considerable competition for other materials. The breakthrough tests are preceded by a careful characterization of the studied materials. Multiple techniques give a quite comprehensive knowledge of their structure and porosity. Especially, the porosity characterization is a demanding task for the pores of such a small size. Taking these advantages into account I would recommend the manuscript for publication, but the authors did not avoid a few inaccuracies or mistakes that should be corrected.

We thank the reviewer for the excellent suggestions and critical examination, that greatly improved the quality of this manuscript after revision. We corrected the inaccuracies or mistakes accordingly and addressed the remarks point-by-point as follows.

1. I should point out that further criticism of the terminology concerns the scientific community rather than authors who have had to succumb to common trends. Beginning with the title, the requirement of self-advertising and drawing attention negatively affects the clarity and precision of the language. The word "hidden" does not seem proper to describe the pores. The inability to detect them by the N₂ sorption does not make them hidden. The authors prove by themselves that the pores are not hidden, but just require more sophisticated approach to characterize them.

Response: Thanks for reviewer's correction. Accordingly, to avoid the irregularity of the wording for describing pores, we have corrected "Hidden Narrow Ultramicrochannels" to "Sub-5Å Micropores" in the title and the main text accordingly.

2. Another inaccuracy is using "ultramicrochannels" to describe the pores. First of all, there is no premises presented to determine that the pores are channels (i.e. they are roughly cylindrically shaped). Moreover, "ultra" is not a strict definition. According to the most popular IUPAC classification micropores have the aperture smaller than 2.0 nm (often defined in the range of 0.3 nm to 2.0 nm). But "ultramicro" (even though a few times used in the scientific literature) has no clear meaning. This resembles exaggerating suffixes in advertisements (like "hyper" or "super").

Response: Thanks for reviewer's corrections. Accordingly, we have changed the "channel" to "pore" in the revised manuscript. Moreover, considering that we specifically focus on the small micropores below 5 Å, we corrected the "ultramicrochannel" to "Sub-5Å Micropores" in the title or "small micropores below 5 Å" in the main text.

3. Finally, I take this opportunity to provide a comment for general consideration (not necessarily changes to the text). Topics, methods, materials etc. are often referred to as "new" or "novel" in modern scientific papers. This is not found in older papers, while their contribution to the progress of science is unquestionable. Do we have to emphasize importance of our works so directly?

Response: Thanks for reviewer's comment. We apologize for the delivery of emphasizing the importance of our work directly. Accordingly, we have removed the word of "new" or "novel" from the Abstract and the last paragraph of Introduction. We agree and appreciate older papers and classic papers that have made great contributions to the progress of science, and we wish to make further contributions of probing and manipulation of sub-5Å in carbonaceous materials and unveiling their efficiency for size-sieving of olefins/paraffins in this work, to the fields of adsorptive separation and materials science.

4. Except these few remarks of general nature I would like I would like to draw attention to the more significant inaccuracies. Interpreting the PALS results is unlike any approach to analyzing such data that I have seen before. Although Zgardzińska recently promotes presenting data in similar way (INTI plots), fig.2d definitely does not present lifetimes and intensities of free positrons, because there should

be a single intensity and lifetime for each sample. The positron lifetime is a parameter that describes the rate of an exponential decay (like in radioactive decays). There are usually several (rarely one) components in a PAL spectrum, so describing the x-axis in Fig.2d as "Lifetime" is hardly correct if the plot consists of dozens of points. In turn, FWHM is reserved for the spectrometer's time resolution function and cannot be dimensionless, so the inset also needs to be revised. In fact, Fig.2d resembles PAL spectra plotted in an uncommon lin-log scale (x-axis should be described as "Time" in such case). In other papers spectra are usually shown in log-lin scale, but most often not shown at all due to their visual similarity. Only numerical analysis with the use of dedicated software (PALSfit, LT, MELT) allows to determine component intensities I_i , lifetimes τ_i and their distributions σ_i (i enumerates components). The results are discussed on this basis. In "Methods. Positron annihilation lifetime spectroscopy experiment." the authors describe using LT9, but I do not see any results of such analysis. The "Methods.

Response: Thanks for reviewer's correction. We apologize for the rush analysis of PALS results. Accordingly, we re-carried out the PALS measurement and analyses more professionally with the assistance from the State Key Lab of Particle Detection and Electronics at USTC. The obtained PALS spectra were deconvoluted into three components using the LTV9 program. The lifetimes and their corresponding intensities of the samples can be repeated within errors. The results were presented in the new Table S2 (also attached below):

Table S2. Average positron-electron lifetime (τ_1 , τ_2 , τ_3) and their corresponding intensities of various samples. The standard deviations for all samples are based on the nonlinear least squares fitting process carried out using the LTV9 software.

Samples	τ_1 (ps)	I_1 (%)	τ_2 (ps)	I_2 (%)	τ_3 (ns)	I_3 (%)	Fitting variance
PDA-C300	167.3 ± 2.3	18.46 ± 0.14	375.2 ± 4.8	80.73 ± 0.14	1.74 ± 0.14	0.82 ± 0.02	1.0036
PDA-C500	180.7 ± 2.8	18.80 ± 0.42	388.4 ± 8.4	81.12 ± 0.42	5.90 ± 1.70	0.08 ± 0.01	1.0363
PDA-C700	179.9 ± 6.2	18.60 ± 0.91	388.5 ± 2.0	81.10 ± 0.91	2.48 ± 0.20	0.30 ± 0.03	0.9928
PDA-C800	168.7 ± 7.6	12.01 ± 0.56	399.7 ± 1.2	87.99 ± 0.56	--	--	1.0332
PDA-C900	164.8 ± 8.4	11.12 ± 0.45	399.6 ± 1.0	88.88 ± 0.45	--	--	1.0462

The measurement and analysis method of PALS has been revised according:

Page 16, line 421-436: PALS was performed to analyze the free-volume of the samples. The positron source (^{22}Na) with an activity of $30 \mu\text{Ci}$ was used and sandwiched between two same disk-shaped pieces. The sample-source-sample set was kept at a room temperature in a vacuum chamber (vacuum better than 1×10^{-3} Pa) during the PALS measurements. One gamma detector is regarded as "start" to detect the 1.275 MeV gamma from the ^{22}Na nucleus that is emitted simultaneously with the beta decay positron.

Another gamma detector is “stop” for detecting one of the two subsequent 0.511 MeV annihilation photons. Electronics measure the “start” - “stop” time intervals between two γ rays, and then lots of annihilation events were recorded for a positron lifetime spectrum for eventual computer fitting. Each lifetime spectrum contains 2×10^6 accumulated counts. The time resolution of the digital PALS spectrometer (TechnoAP, Japan) was 190 picosecond (ps) and the count rate was around 200 cps. The PALS spectra were analyzed using the LTV9 program.

In our work, the free-volume size was determined by using a semi-empirical infinite potential spherical model, where the average diameter (2R) of nanoscale free-volume cavities could be estimated from the positron lifetime τ_2 :³⁰

$$\tau_2 = 0.260 \times \left[1 - \frac{R}{R+3.823} + \frac{1}{2\pi} \sin\left(\frac{2\pi R}{R+3.823}\right) \right]^{-1} \quad (1)$$

where, the units of τ_2 and R are expressed in ns and Å, respectively.

From the analyses, the average sizes of free-volume holes of PDA-Cx were shown as the new Fig. 2c.

Fig. 2c The average pore size in PDA-Cx derived from the PALS.

Turned out the PALS results were interesting and helped us build a deeper understand the PDA-Cx materials. As can be seen in Fig. 2c, PALS-detected average diameter of free-volume cavities increased from 5.13 Å of PDA-C300 to 5.63 Å of PDA-C900, revealing the formation of slightly larger angstrom-sized voids in carbon matrix induced by higher pyrolysis temperature. However, from the gas probing tests, negligible N_2 uptake at 77 K (<0.1 mmol/g) but significant amounts of H_2 at 77 K and CO_2 at 195 K, were detected in PDA-Cx. Above phenomena suggest that smaller micropores below 5 Å (accessible for the smaller H_2 and CO_2 probes, but not N_2 probe) in PDA-Cx are present as the orifice [Carbon, 1969,7(6), 643-648; Carbon, 1970, 8(3), 353-364], to limit N_2 to diffuse at critical temperature to the larger microcavities (5.13-5.63 Å) detected from PALS spectra, but still accessible for the smaller H_2 and CO_2 probes. Therefore, it can be understood as that the small micropore orifices in PDA-C800 and PDA-C900 distribute narrowly with a size deviation (defined as pore size distribution at maximal full-width) of 0.2-0.3 Å, which is less than the size discrepancy of the C_2/C_3 olefin/paraffin pairs, allowing the entrance of olefins while entirely exclude larger paraffin counterparts, while the larger microcavities allow high C_2H_4 and C_3H_6 uptakes of 2.25 and 1.98 mmol/g at 298 K and 1.0 bar, respectively. Ideal

molecular recognition behavior has been further demonstrated by binary breakthrough experiments. Based on the above results, we've revised the corresponding parts throughout the MS.

Page 1, Line 24-30: ".....we reported polydopamine-derived carbons (PDA-Cx) with tailorable sub-5 Å micropores orifices together with hyper-5 Å microcavities by one-step pyrolysis. The sub-5 Å micropores orifices centered at 4.1-4.3 Å in PDA-C800 and 3.7-4.0 Å in PDA-C900 allow the entry of olefins while entirely exclude their paraffin counterparts, performing a precise cut-off to discriminate olefins/paraffins with sub-angstrom discrepancy. The hyper-5 Å microcavities enable high C₂H₄ and C₃H₆ capacity of 2.25 and 1.98 mmol/g under ambient conditions."

Page 6, Line 139-149: "The above PALS technique reflects the variation of average pore size of the accessible free-volume in porous frameworks³¹, whereas the probe gases can estimate the size of pore orifices accurately¹⁰. To further unveil the size distribution of pore orifices, N₂ physisorption at 77 K was performed. Negligible N₂ uptake (<0.1 mmol/g) was detected in PDA-Cx at 1.0 bar except for PDA-C700 (Supplementary Fig. 10). However, by further applying probe gases of H₂ and CO₂, PDA-Cx exhibited remarkably higher H₂ and CO₂ uptakes at higher pyrolysis temperature, where PDA-C900 realized 5.28 mmol/g for H₂ capture at 77 K and 1.0 bar, and 5.60 mmol/g for CO₂ capture at 195 K and 1.0 bar (Supplementary Figs. 11-12). Above phenomena suggests that smaller sub-5 Å micropores are present in PDA-Cx as the orifice^{32,33}, to limit N₂ to diffuse at critical temperature to the larger microcavities (5.13-5.63 Å) detected from PALS spectra, but still accessible for the smaller H₂ and CO₂ probes....."

Page 12, Line 295-296: ".....From thermal-controlled pyrolysis of polydopamine, PDA-Cx possess refined sub-5 Å small micropore orifices together with 5.13-5.63 Å microcavities."

Page 12, Line 301-303: ".....allowing the entrance of olefins while entirely exclude larger paraffin counterparts. The larger microcavities allow high C₂H₄ and C₃H₆ uptakes of 2.25 and 1.98 mmol/g at 298 K and 1.0 bar, respectively."

5. Positron annihilation lifetime spectroscopy experiment." part also require some improvement:

5-1 Description "electronics" is not sufficient for PAL spectrometer.

Response: Thanks for reviewer's correction. In the revised manuscript, we added the description as "One gamma detector is regarded as "start" to detect the 1.275 MeV gamma from the ²²Na nucleus that is emitted simultaneously with the beta decay positron. Another gamma detector is "stop" for detecting one of the two subsequent 0.511 MeV annihilation photons. Electronics measure the "start" - "stop" time intervals between two γ rays, and then lots of annihilation events were recorded for a positron lifetime spectrum for eventual computer fitting. Each lifetime spectrum contains 2×10⁶ accumulated counts. The time resolution of the digital PALS spectrometer (TechnoAP, Japan) was 190 picosecond (ps), and the count rate was around 200 cps. The PALS spectra were analyzed using the LTv9 program." (Page 16, line 424-431)

5-2 I cannot agree with the statement: "the value of τ_3 is low due to the weak electronic conductivity of the pyrolyzed samples". Positronium (usually the third component) is not formed (or quenched with high rate) in conductive samples, not the other way around, as the authors write.

Response: Thanks for reviewer's correction. Accordingly, the statement has been corrected to "The second-component positron lifetime (τ_2) was used to estimate the average pore size (Equation 1), as nearly no o-Ps component (τ_3) was detected in PALS due to the electronic conductivity of PDA-Cx³⁰." in the Page 5, line 129-131.

5-3 What is (s3) described as the component? The symbol is the most similar to σ_3 , which is the distribution of lifetimes.

Response: We thank the reviewer for pointing out the inaccuracy. The "s3" described in the third long-lived component should be " σ_3 ". The error was corrected accordingly. In the revised discussion of PALS, we did not mention the distribution of o-Ps component (σ_3) as no o-Ps component is observed in PALS results (Table S2). The free-volume data from PALS was analyzed based on the second-component positron lifetime (τ_2) in our work.

5.4 PALS results cannot be accepted in the present form. However, if fig. 2d presents the spectra, even their visual differences allow to expect that interesting and valuable information can be obtained from PALS. Therefore, I encourage the authors to analyze the PALS data properly and draw conclusions from the results.

Response: Thanks for reviewer's correction. PALS data has been analyzed properly based on a semi-empirical infinite potential spherical model (Y. C. Jean, et al. *Macromolecules*, **2011**, 44(17), 6818-6826), and the sizes of free-volume cavities of PDA-Cx were shown in the new Fig. 2c.

Fig. 2c The average pore size of PDA-Cx derived from the PALS.

To clarify, the discussion of results in the revised manuscript regarding to the PALS has been revised accordingly in Page 5, line 126-133 the revised manuscript: "Positron annihilation lifetime spectroscopy (PALS) is a useful microprobe capable of direct determining the local void spaces and average pore size in a 3D network at the atomic scale²⁹. The detailed positron lifetime components (τ_1 , τ_2 , τ_3) and their corresponding intensities were listed in the supplementary Table 2. The second-component positron lifetime (τ_2) was used to estimate the average pore size (Equation 1), as nearly no o-Ps component (τ_3) was detected in PALS due to the electronic conductivity of PDA-Cx³⁰. PALS-detected

average pore size increased from 5.13 Å of PDA-C300 to 5.63 Å of PDA-C900, revealing the formation of slightly larger angstrom-sized voids in carbon matrix induced by higher pyrolysis temperature (Fig. 2c).”

6. It is not clear what authors mean by limitation of diffusion in description of N₂ sorption. In addition to diffusion, adsorption involves several other phenomena. Here, I would expect that not only quite arbitrary defined molecule sizes, but also the energies of the adsorbate-adsorbent adsorbate-adsorbate interaction play important role. Probably limiting it to van der Waals forces is also a simplification. As the authors correctly write, simple adsorption models do not work for these pore sizes and the example of CO₂ adsorption at different temperatures confirm the importance of interaction energy. Therefore, reducing the discussion of adsorption to steric restrictions as Fig. S13 is a simplified interpretation (even though it is convenient and easy to understand). This should be clearly written to avoid misleading a reader. Neutron results shed some light on this, but they are presented many paragraphs later. Probably they should be at least mentioned or briefly announced in discussion of N₂ sorption.

Response: Thanks for reviewer’s comment. We agree with the reviewer that apart from steric restrictions, the interaction energy of adsorbent-adsorbate and adsorbate-adsorbate may play an important role, therefore it is necessary to discuss it before focusing on steric restrictions. To clarify, the parenthetical text below has been added to the main body.

Page 6-7, line 154-160 “..... Additionally, it was noted that the CO₂ adsorption of PDA-Cx at 273 K exhibited less capacity increase at low pressure compared with that at 195 K (Supplementary Figs. 12 and 15), revealing the weak host-guest interaction between CO₂ molecules and non-polar carbon surfaces by van de Waals forces. The weak physisorption can be further confirmed by Inelastic neutron scattering in later part. The adsorption is thus reduced to steric restrictions to probe the pore size by various gas molecules with the sub-angstrom size discrepancy as a simplified interpretation.”

7. The size deviation of the PDA-C800 and PDA-C900 pores (estimated by Dubinin-Astakhov equation), AC-1 and CMS-1 (from N₂ sorption) is not clearly defined. It does not seem to be a standard deviation or FWHM. Moreover, comparing pore sizes and their distributions obtained by different methods for these samples is not proper. Especially keeping in mind previously presented failure to determine pore sizes for PUC with N₂. I wonder why the same adsorption probes as for PUC were not used for AC-1 and CMS-1 – comparison of such results for small pores would be interesting and would allow to verify PUC results.

Response: Thanks for reviewer’s suggestions. Accordingly, we defined “size deviation” as “pore size distribution at maximal full-width”, and added the definition in the main text (Page 7, Line 176) for clarification in the revised MS. Moreover, as the reviewer suggested, we carried out additional molecular probing experiments for AC-1 and CMS-1, and analyzed the small micropores below 5 Å on AC-1 and CMS-1 by using a series of probing gases based on the Dubinin-Astakhov equation. The results are presented in the new Figs. S21-22 and 24, and the following discussion are made and added to the main body (also attached below):

Page 7, line 178-183 “..... In distinct contrast to the commercial carbons, the major porosities of AC-1 were broadly distributed in the micro-mesopore region ranging of 5.8-31.4 Å, while that of CMS-1 were more narrowly distributed in micropore region of 5.8-15.4 Å, according to N₂ physisorption at 77 K (Supplementary Fig. 23). Meanwhile, bare small micropores below 5 Å were probed in CMS-1, and a small portion of small micropores but below 3.7 Å were probed in AC-1 (Supplementary Fig. 24).”

Figure S21. Probe molecular adsorption isotherms on (a) AC-1 and (b) CMS-1 at 273 K.

Figure S22. Typical D-A plot of probe molecules on (a) AC-1 and (b) CMS-1 at 273 K.

Figure S24. a) The pore volumes of AC-1 and CMS-1 calculated from different probe gases based on the Dubinin-Astakhov equation (The data points from left to right are calculated from probing gases of CO₂, Ar, C₂H₄, C₂H₆, C₃H₆, C₃H₈, CF₄, and i-C₄H₁₀); b) The corresponding pore size distributions. (The differential pore volume (ΔV) was obtained from probes with successive sizes. V_{Ar} calculated by Ar was subtracted from V_{CO_2} , $V_{C_2H_4}$ from V_{Ar} , and so on. The last one $V_{i-C_4H_{10}}$ was pore volume with a size larger than 5.0 Å.

8. I could not find any explanation how the pore size distributions presented in Fig.3d and S18b were calculated.

Response: Thanks for the referee's comment. In our work, the pore size distribution is calculated by the differential pore volume obtained from probe gases with different molecule sizes based on the Dubinin-Astakhov equation. For example, the molecule size of CO₂ and Ar is 3.3 Å and 3.5 Å, respectively. The pore volume of PDA-C900 calculated from CO₂ (V_{CO_2}) is 0.21 cm³/g, and the pore volume calculated from Ar (V_{Ar}) is 0.20 cm³/g. V_{Ar} was subtracted from V_{CO_2} , and then the differential pore volume (ΔV) with a size falling in-between 3.3-3.5 Å is 0.01 cm³/g. Similarly, the pore size distribution in the range of small micropore region (3.3-5.0 Å) can be obtained.

To clarify, the calculation method of pore size was added in the revised manuscript:

Page 17 and line 455-460:where V_0 is the micropore volume (cm³/g) and ρ_M is the liquid molar density (mmol/cm³). Thus, combining equation (1)(2)(3), a series of micropore volume (V_0) calculated from different gas probes can be obtained. After that, the V_0 calculated from gas A was subtracted from V_0 calculated from gas B. In this way, the differential pore volumes can be obtained, as shown below:

$$\Delta V = V_{0(\text{gas A})} - V_{0(\text{gas B})} \quad (5)$$

Where, ΔV is the pore volume with a size falling in-between the molecular size of gas A and gas B.

For the calculation of pore volume, please find the detailed equations in Page 17, line 444-455:

D-A equation is formally written as⁴⁹:

$$W = W_0 \exp\left[-\left(\frac{A}{\beta E_0}\right)^n\right] \quad (2)$$

Here, W refers to the adsorbed capacity (mmol/g), W_0 is the saturation capacity (mmol/g), β is termed the similarity affinity coefficient of the characteristic curves, E_0 is the micropore characteristic energy of adsorption, and n is an empirical constant related to surface heterogeneity. If $n=2$, the D-A equation reduces to the D-R form. A is the adsorption potential by Polanyi, that can be defined as:

$$A = RT \ln\left(\frac{P_0}{P}\right) \quad (3)$$

Where, P_0 and P represent saturated vapor pressure and adsorption pressure (bar), respectively. T is the temperature (K), R is the gas constant. Therefore, W_0 can be calculated from the intercept of the line plot of $\ln W$ vs $\ln^n (P_0/P)$. Furthermore, W_0 also defined as:

$$W_0 = V_0 \rho_M \quad (4)$$

where V_0 is the micropore volume (cm^3/g) and ρ_M is the liquid molar density (mmol/cm^3).

9. Mercury porosimetry is suitable to find macropores not mesopores. Moreover, it raises suspicions if high pressure influences measured values. Especially in the range of smaller pores.

Response: Thanks for reviewer's correction. We agree with the reviewer that mercury porosimetry can only provide pore information of macropores, rather than mesopores and smaller micropores, as it remains uncertainty if high pressure influences measured values in the range of smaller pores. Therefore, we have removed the data of pore sizes below <50 nm from mercury porosimetry in the revised Fig. S25 (also attached below). Instead, the smaller pores in our work were probed under ambient pressure by N_2 physisorption at 77 K and various gas probes with the sub-angstrom size difference.

Figure S25. Macroporosity fraction of PDA and PDA-Cx obtained from mercury porosimetry.

10. The description of the results in the chapter "Dynamic separation of olefins/paraffins", i.e. 80% and equilibrium times cannot be read from fig. S37 and S38. What "Sqt" in this figure means? Maybe this was meant "sqrt", "square root"? In such case I suggest to present data on the square root scale (or log scale), where time is given in minutes, not in square root of minutes. Slightly increasing value in fig.

S37 does not support the sentence "No detectable C₂H₆ uptake was found within the whole 100 min" (and 100 min confirms a sqrt scale here). This is a little confusing. What are uncertainties of the calculated diffusion rates? They can be easily calculated from fitting.

Response: We thank the reviewer's suggestion for improving the figure. To make the critical data more readable, (a) We re-carried out the kinetic curves and marked the "80% and equilibrium time" in the Fig. S37 and S38 (new Fig. 5a in the revised manuscript) by using the dotted line. (b) Yes, "Sqrt" should be "Sqrt" and means "square root". To avoid the confusion for readers, the x-axis of kinetic curves in Fig. 5a has been corrected to "Time (min)" directly, and only the x-axis of Fig. S41 was "Time^{1/2} (min^{1/2})", where the adsorbed quantity, q_t/q_∞ , varies linearly as a function of the square root of time according to the equation (6) in the main text. The sentence "No detectable C₂H₆ uptake was found within the whole 100 min" has been corrected to "Bare C₂H₆ adsorption (<0.01 mmol/g) was found within the whole 100 min" accordingly. (c) The uncertainties of the calculated diffusion rates from fitting were added in the new Fig. S41 (also attached below).

Figure 5a. Time-dependent gas uptake profiles of C₃H₆ on PDA-C800 and C₂H₄ on PDA-C900. The time required to reach 80% of the saturated olefin uptake and equilibrium is depicted as dashed lines.

Figure S41. The diffusional time constant calculation details for a) C₃H₆ in PDA-C800 and a) C₂H₄ in PDA-C900.

11. The description of models in "Methods. Molecular probe calculation" does not lead to $\Delta V(D)$, which is presented in figures. I suggest to extend this description to show the full course of calculations.

Response: We thank the reviewer for the suggestion. Accordingly, we have added the following explanation of detailed calculation of ΔV in the description of models in "Methods. Molecular probe calculation", to show the full course of calculations, in the revised manuscript.

Page 17 and line 455-460:where V_0 is the micropore volume (cm^3/g) and ρ_M is the liquid molar density (mmol/cm^3). Thus, combining equation (1)(2)(3), a series of micropore volume (V_0) calculated from different gas probes can be obtained. After that, the V_0 calculated from gas A was subtracted from V_0 calculated from gas B. In this way, the differential pore volumes can be obtained, as shown below:

$$\Delta V = V_{0(\text{gas A})} - V_{0(\text{gas B})} \quad (5)$$

Where, ΔV is the pore volume with a size falling in-between the molecular size of gas A and gas B.

12. What are n values in fig. S17?

Response: We thank the reviewer for the question. In Fig. S17, n is an empirical constant related to surface heterogeneity, that is one of the important parameters in empirical Dubinin-Astakhov (D-A) and Dubinin-Radushkevich (D-R) equations. D-R equation is considered more accurate for homogeneous micropore surfaces, where n is taken to be 2. Differently, the modified D-A equation is developed to determine the adsorption isotherms on surfaces that have some degree of heterogeneity, thus that is more general and reliable for carbon materials with non-crystalline structure (*Langmuir* 1998, 14, 3840-3846). The value of n in D-A equation is determined by fitting the model against experimental data.

Accordingly, the explanation of n in the main body can be seen in the Page 17, line 448-449. Meanwhile, the fitting parameters of the D-A equation (including the value of " n ") were added in the Supplementary Tables S4-6:

Table S4. Fitting parameters of the D-A equation on high temperature pyrolyzed PDA-Cx.

Gases	PDA-C900			PDA-C800			PDA-C700		
	$\ln W_0$	n	R^2	$\ln W_0$	n	R^2	$\ln W_0$	n	R^2
CO ₂	1.70	2.10	0.9998	1.61	2.25	0.9991	1.46	2.42	0.9998
Ar	1.94	1.60	0.9999	1.87	1.46	0.9999	1.72	1.69	0.9999
C ₂ H ₄	1.28	2.16	0.9951	1.26	2.25	0.9961	1.07	2.62	0.9941
C ₂ H ₆	-1.36	1.76	0.9936	0.94	1.83	0.9967	0.86	2.48	0.9992
C ₃ H ₆	-0.81	1.27	0.9433	0.85	1.44	0.9952	0.76	2.39	0.9932
C ₃ H ₈	-1.93	1.19	0.9958	-1.24	1.41	0.9904	0.46	0.65	0.9536
CF ₄	-1.53	2.12	0.9991	-1.88	2.25	0.9902	-1.62	1.80	0.9901
i-C ₄ H ₁₀	-1.96	1.29	0.9912	-1.99	1.42	0.9883	-1.79	1.25	0.9901

Table S5. Fitting parameters of the D-A equation on low temperature pyrolyzed PDA-Cx.

Gases	PDA-C500			PDA-C300		
	lnW ₀	n	R ²	lnW ₀	n	R ²
CO ₂	1.07	2.40	0.9999	0.33	2.20	0.9999
Ar	1.32	1.37	0.9996	0.55	1.50	0.9900
C ₂ H ₄	0.65	2.80	0.9993	-0.18	1.26	0.9955
C ₂ H ₆	0.50	2.63	0.9985	-1.21	1.90	0.9932
C ₃ H ₆	--	--	--	--	--	--
C ₃ H ₈	--	--	--	--	--	--
CF ₄	-1.15	2.02	0.9988	-3.35	2.65	0.9614
i-C ₄ H ₁₀	--	--	--	--	--	--

Table S6. Fitting parameters of the D-A equation on AC-1 and CMS-1.

Gases	AC-1			CMS-1		
	lnW ₀	n	R ²	lnW ₀	n	R ²
CO ₂	3.35	1.42	0.9999	1.70	1.97	0.9999
Ar	3.48	1.49	0.9998	1.89	1.27	0.9999
C ₂ H ₄	2.78	1.75	0.9999	1.27	2.27	0.9998
C ₂ H ₆	2.66	1.87	0.9998	1.12	2.26	0.9992
C ₃ H ₆	2.44	1.95	0.9997	0.98	2.41	0.9981
C ₃ H ₈	2.34	2.03	0.9994	0.83	2.12	0.9983
CF ₄	2.70	1.35	0.9994	1.16	1.62	0.9971
i-C ₄ H ₁₀	2.14	2.53	0.9911	0.65	1.13	0.9969

13. Last part of my comments concern clarity and ease of reading of the manuscript.

13-1. Using the Supplementary Materials is not convenient. A reader has to read both documents simultaneously to understand the text. Either some of the discussion referring to figures in SI should be moved to SI and only simple conclusion should be left in main text or the figures from SI should be moved to the main text.

Response: Thanks for reviewer's nice suggestions. Accordingly, we made the following revisions:

- 1) Important figures have been moved from SI to the main text, including kinetic curves of olefins and paraffins (old Fig. S37 and 38, new Fig. 5a), and the heat of adsorption (old Fig. 46, new Fig 5f).

2) Minor discussion referring to the figures in SI have been moved to SI and only a concise conclusion was kept in the main text, including:

1. In the main text, the discussion with FT-IR, XPS and solid NMR (Page 5-6, line 109-116) was reduced to “To track the structural change of heteroatom bonding nature during pyrolysis, X-ray photoelectron spectra (XPS) was utilized, which indicates the transformation from pyrrolic-N to pyridinic-N and thermally stable graphitic N, existing at edges and defects sites (Supplementary Figs. 4-7 and Table 1). ¹³C NMR spectra validates the loss of original aliphatic CH_x groups (x=1~3) and functional groups (O-H/N-H) (Fig. 2a). A rather graphitic carbon phase formed, although the signal intensity of aromatic rings reduced due to the lack of hydrogen at the sites²⁵, consistent with Fourier transform infrared (FT-IR) spectra (Supplementary Fig. 8).”

2. The experimental details of SAXS and WAXS were moved to Supplemental materials, and a concise conclusion was kept in the Page 7-8, line 186-189 in the main text: “.....Furthermore, from the mercury porosimetry and small-angle X-ray scattering, a very small amount of macropores (~1 μm) in PDA-Cx are detected, which can be related to the interparticular voids that barely contributes to the adsorption of angstrom-size gases (Supplementary Figs. 25-26).”

3. We left the calculation method of pore size in the main text (Page 16-17, line 437-460), and moved the equation of evaluating saturated vapor pressure (P₀) of the probe gases to the SI (at the end of Table S3).

4. The result of Raman spectra was moved to supporting information (old Fig. 2c, new Fig. S9).

3) To be more readable, the fitting parameter of the heat of adsorption was now listed in the new Table S11 in the revised Supplemental material, instead of in the previous Fig. S45, which was not easy to read directly.

Table S11. Fitting parameters of the virial equation and the corresponding correlation coefficients.

	a_0	a_1	a_2	a_3	a_4	R^2
C ₂ H ₄	-3890.26	16.6033	0.28916	-0.00445	2.52E-05	0.9985
C ₃ H ₆	-2981.62	-35.158	0.26814	0.00863	-7.28E-5	0.9973

13-2. The structural formulas in fig.2a are very small.

Response: Thanks for reviewer’s remark. Accordingly, we have enlarged the structural formulas in the Fig. 2a to make it clearer. Meanwhile, we thoroughly proofread the whole manuscript and corrected such unclarities.

13-3. Reading comparison of the sorption of olefins/paraffins I miss the figure with comparison of the pore sizes of all PDA-Cx made by the same method (from the Dubinin-Astakhov equation or a little

mysterious calculation based on the size of gas probes). Even these presented for PUC-700, PDA-C800 and PDA-C900 are divided to main text and the Supplementary Materials.

Response: Thanks for reviewer’s comment. Accordingly, we carried out experiments and analyzed the pore size distribution of PDA-C700 as well as PDA-C300 and PDA-C500, and now included all the series of data, including the comparison of pore volume and pore size distribution in the new Fig. 3c and 3d in the revised MS, with the calculation details in Figs. S17-20.

In addition, the parenthetical below has been added in the revised manuscript:

Page 7, line 170-176: “.....Moreover, the difference in micropore volume can be reduced to sterical restrictions in pores smaller than the size of probes, resulting in a high-resolution PSD below 5 Å (Equation 4). Apart from PDA-C300 with low porosity, the small micropore orifice size of PDA-C500 was determined to be falling in-between 4.1-4.7 Å. Higher temperature contributes to the evolution of micropores towards smaller size driven by thermal energy. Noticeably, small pore sizes of PDA-C800 and PDA-C900 fell in-between 4.1-4.3 Å and 3.7-4.0 Å, respectively, with a size deviation (defined as pore size distribution at maximal full-width) of 0.2-0.3 Å (Fig. 3d).”

Fig. 3 Sorption behavior of gas probes and pore information. Sorption equilibrium isotherms of various gas probes with minimum molecular dimensions ranging from 3.3 Å (CO₂) to 5.0 Å (i-C₄H₁₀) at 273 K and N₂ at 77 K on **a** PDA-C800 and **b** PDA-C900 materials; **c** The pore volumes of PDA-Cx calculated from different probe gases based on the Dubinin-Astakhov (D-A) equation (the data points from left to right were calculated from probing gases of CO₂, Ar, C₂H₄, C₂H₆, C₃H₆, C₃H₈, CF₄, and i-C₄H₁₀). Among them, C₃H₆, C₃H₈, and i-C₄H₁₀ with high polarizability were not applied to heteroatom-rich PDA-C300 and PDA-C500 due to the larger impact of host-guest interaction); **d** The pore size distributions of PDA-Cx (the differential pore volume (ΔV) of y-axis was obtained from probes with successive sizes. V_{Ar} calculated by Ar was subtracted from V_{CO₂}, V_{C₂H₄} from V_{Ar}, and so on).

Figure S17. a) Probe molecular adsorption isotherms and b) typical D-A plot on PDA-C300 at 273 K.

Figure S18. a) Probe molecular adsorption isotherms and b) typical D-A plot on PDA-C500 at 273 K.

Figure S19. a) Probe molecular adsorption isotherms and b) typical D-A plot on PDA-C700 at 273 K.

13-4. I suggest to enlarge figures in Supplementary Materials where possible-currently high magnifications of the document are required to see their details.

Response: We thank the reviewer’s suggestion. Accordingly, we have enlarged figures and increased its resolution in the Supplementary Materials.

13-5. There are no labels above the break of y-axis in fig. S34, which does not allow to read values on this figure without looking at fig. 4c.

Response: Thanks for the reviewer’s comment. For size-sieving adsorbents, since the paraffin capacity is extremely low, ideally approaches zero, and thus the olefin/paraffin uptake ratio can be rather inaccurate and meaningless (generally higher than 20) or IAST selectivity larger than 100 [*J. Am. Chem. Soc.* **2021**, 143, 46, 19300–19305], and ideally would be approaching infinity. Therefore, we added the label of “size-sieving adsorbents” region to classify them in the Fig. S34 (re-numbered Fig. S38). And referred to the literature reporting size-sieving adsorbents in supplementary Tables S7-8, we marked the olefin/paraffin uptake ratio of 20 as a bar to distinguish the sieving adsorbents from the others, and added the explanation in the figure caption of Fig. S38 as “**The size-sieving adsorbents are marked in yellow (a) and green (b) regions with olefin/paraffin uptake ratio over 20**”

Figure S38. Comparison of olefin adsorption capacity and olefin/paraffin separation factor of PDA-C800 and PDA-C900 developed in this work with state-of-the-art porous adsorbents reported in the literatures. The dotted line is the current limitation of capacity and separation factor. The details are given in supplementary Tables S7-8. The size-sieving adsorbents are marked in yellow (a) and green (b) regions with olefin/paraffin uptake ratio over 20.

Reviewer: 3

Comments:

The authors present a study on the ultramicropores of carbons that are not easy to find or access. Therefore, they prepare materials from pre-ordered polymeric material. Many measurements were performed to investigate the nature of those pores, ranging from gas sorption with species such as CO_2 and O_2 up to bigger molecules, but also synchrotron methods were used such SAXS, WAXS and positron annihilation lifetime spectroscopy (PALS). The found pore-sizes in the range of 3-5 Å seemed to be interesting for olefin paraffin separations, especially focusing on propylene/propane separation.

The sorption properties were probed, and weak physisorption for the probe molecules was found using neutron scattering. Therefore, the authors apply a breakthrough analyzer to determine separation performance. Some questions arise from reading the manuscript:

We thank the reviewer for the careful evaluation and suggestions.

1. In your introduction you mention the reference 24, which has proven the hidden micropores. You are adapting their method but are not able to reproduce it for your samples correctly. Somehow, I am missing the part where you are mentioning the pore volume for the micropores in the paper. Table S2 has them, but they are not referred to in the main text – because they are super low. In the reference-paper, those volumes are also extremely low and this is the reason why these pores are “hidden” – you have to look very exactly at the material. If they are that low in your material, there is no way these channels are responsible for the strong separation values you measure in the end. Please re-evaluate your methods especially for PDA-C900. (see also comment 2)

Response: Thanks for the reviewer’s comment.

We apologize for the missing data of micropore volume of PUC-*x* (not in Table S2), and now added them in the Fig. 3c (also attached below) in the revised MS. Results show that the micropore volume of PUC-900 (re-named PDA-C900) based on CO₂ probe reaches up to 0.21 cm³/g, which is actually higher than most of reported size-sieving adsorbents, and thus can be significant for adsorption. These micropores are rather small (<5 Å, named as sub-5Å micropores after revision) to be detectable by the conventional N₂ probe at 77 K (also referred to reference 24), and this is the reason why these pores are “hidden”. Regardless, the pore sizes of these sub-5Å micropores precisely fall in-between the molecule sizes of C₂- and C₃- olefin and paraffin counterparts, as probed by a series of size-differentiating probing gases, thus performing the strong separation values we measured in the end via size-sieving effect.

Fig. 3. c Pore volume of PDA-C_x calculated from a series of probing gases with the minimum molecular dimension ranging from 3.3-5.0 Å based on the Dubinin-Astakhov (D-A) equation (the data points from left to right are calculated from probing gases of CO₂, Ar, C₂H₄, C₃H₆, C₂H₆, C₃H₈, CF₄, and i-C₄H₁₀. Among them, C₃H₆, C₃H₈, and i-C₄H₁₀ with high polarizability were not applied to heteroatom-rich PDA-C300 and PDA-C500 due to the larger impact of host-guest interaction).

Please see the detailed explanation below:

1) Pore volume calculation

The pore volume of PDA-Cx is now presented in Fig. 3c in the revised MS. The calculation is based on the Dubinin-Astakhov (D-A) equation, which is a traditional and reliable method to determine the micropore volume of porous materials (*Langmuir*, 1998, 14(14), 3840-3846; *Chem. Eng. Sci.*, 2003, 58(14), 3059-3075). And the pore size is calculated based on the differential pore volume obtained from various probe gases. Larger molecules, due to their size, are partially excluded from the pores resulting in a lower pore volume. For example, the molecule size of CO₂ and Ar is 3.3 Å and 3.5 Å, respectively. The pore volume of PDA-C900 calculated from CO₂ (V_{CO_2}) is 0.21 cm³/g, and the pore volume calculated from Ar (V_{Ar}) is 0.20 cm³/g. V_{Ar} was subtracted from V_{CO_2} , and then the pore volume with a size falling in-between 3.3-3.5 Å is 0.01 cm³/g. Similarly, the pore size distribution in the range of hidden micropore region (3.3-5.0 Å) can be obtained (the detailed calculation method can be found in Page 16-17, line 437-460 in the revised MS). The information provided in Table S2 (new Table S3) is the physical property of various probes used for calculating pore size.

2) The comparison of pore volume with other materials

As also sensed by the reviewer, when reducing the pore sizes to sub-5Å, the pore volume of carbon can be quite low unavoidably, compared to carbons with larger micro- and meso- pores. Nevertheless, the pore volume of PDA-C900 in our work is still higher (0.21 cm³/g) than most of benchmark size-sieving adsorbents. Table R1 and R2 compare the values of pore volume of C₂- and C₃- olefin/paraffin size-sieving adsorbents, as shown below:

Table R1. The comparison of pore volume of both reported C₂H₄/C₂H₆ size-sieving adsorbents

Materials	Pore volume	References
PDA-C900	0.21 cm³/g	this work
UTSA-280	0.18 cm ³ /g	Nat. Mater. 2018, 17(12), 1128-1133
Co-gallate	0.21 cm ³ /g	Angew. Chem. Int. Ed. , 2018, 130(49), 16252-16257

Table R2. The comparison of pore volume of both reported C₃H₆/C₃H₈ size-sieving adsorbents

Materials	Pore volume	References
PDA-C800	0.18 cm³/g	this work
KAUST-7	0.095 cm ³ /g	Science , 2016, 353(6295), 137-140
Co-gallate	0.21 cm ³ /g	J. Am. Chem. Soc. , 2020, 142(41), 17795-17801
Y-abtc	0.18 cm ³ /g	Adv. Mater. , 2018. 30(49), 1805088
HIAM-301	0.26 cm ³ /g	J. Am. Chem. Soc. 2021.143(46).19300-193005
SCMS-0.2-800	0.24 cm ³ /g	J. Mater. Chem. A , 2021, 9, 23873-23881
C-CDMOF-2-700	0.19 cm ³ /g	ACS Appl. Mater. Interfaces 2022, 14, 26, 30443–30453

3) **The reason for using various probes with sub-angstrom discrepancy for analyzing pores**

We agree with the scientific methodology in ref. 24 (ref. 21 in the revised MS) for probing hidden micropores by H₂ or O₂ adsorption at 77 K. Indeed, it is helpful to confirm the presence of hidden micropores (3.3-5.0 Å) in carbons that are undetectable by N₂. However, to probe the sub-angstrom size discrepancy of such small sub-5Å micropores for separating C₂- and C₃- olefin/paraffin pairs, we applied a series of gas probes with sub-angstrom size discrepancy.

2. In your PDA-Cx you find macropores and tried to proof the “hidden” micropores inside your carbons through various methods, which you then conclude to be the factor for high separation values. Further, you proof that there is not much adsorption happening on your carbons. Nevertheless, you find good separation values for paraffin/olefin separation in your carbon materials. Now, changes in breakthrough are rather influenced by sorption than size-sieving. Thus, if there are lots and lots of larger pores available and no sorption selectivity can be found, why should this material be so good in this type of gas separation? Please explain to me how your data combines rational to the effect you find in breakthrough.

Response: Thanks for the reviewer’s question.

Our ration is: in our PDA-Cx (re-named PDA-Cx in the revised manuscript) we find macropores (~1 μm in size), and the amount is rather small, thus contribute little to the adsorption of olefin/paraffin with molecule size <5 Å (0.0005 μm). Then we proof the significant amount of “hidden” micropores inside our carbons through various methods. The small micropores (<5 Å), rather than the large macropores (~1μm), can yield high adsorption uptakes of these gases (2.25 mmol/g of C₂H₄ for PDA-C900, 1.98 mmol/g of C₃H₆ for PDA-C800) via Van der Waals’ force. Moreover, these sub-5 Å micropores are probed to possess pore sizes narrowly falling in-between C₂- and C₃- olefin and paraffin counterparts, and thus demonstrates high separation values in breakthrough via a size-sieving effect.

We apologize for the misleading writing, to avoid the irregularity of the wording for describing pores, we have corrected “Hidden Narrow Ultramicrochannels” to “Sub-5Å Micropores” in the title and the main text accordingly, and have improved the writing throughout the MS. Also, the sentence of describing macropores is revised to “.....Furthermore, from the mercury porosimetry and small-angle X-ray scattering, a very small amount of macropores (~1 μm) in PDA-Cx are detected, which can be related to the interparticular voids that barely contributes to the adsorption of angstrom-size gases (Supplementary Figs. 25-26).” (Page 7-8, line 186-189)

Please see the detailed explanation below:

1) **The proof of negligible macropores**

According to the result of Mercury Porosimetry (Fig. S25, also attached below), only small amount of macropores existed in PDA-Cx (<4 m²/g), though the total surface area of PDA-C900 is 400 m²/g (Fig. S14). Moreover, the size of macropores in PDA-Cx is ~1 μm, which can be ascribed to the

interparticular voids. Such large macropores ($\sim 1 \mu\text{m}$) have little contribution on the gas adsorption. For example, PDA polymer, as the precursor of PDA-C900, possesses macropores at $0.9 \mu\text{m}$ at the surface area of $7.2 \text{ m}^2/\text{g}$, its C_2H_4 uptake is as low as 0.094 mmol/g at 298 K and 1.0 bar (Fig. S27).

Figure S25. Macroporosity fraction of PDA and PDA-C x samples obtained from mercury porosimetry.

2) The relationship between small sub-5 Å pores and excellent selectivity in breakthrough results

By applying a series of probing gases, the pore size of PDA-C900 is analyzed to be $3.7\text{-}4.0 \text{ \AA}$, which falls in-between the molecule sizes of C_2H_4 and C_2H_6 , and thus enables the size-sieving of C_2H_4 and C_2H_6 in breakthrough experiments (Fig. 5c). Also, PDA-C900 possesses a high C_2H_4 capacity of 2.25 mmol/g , due to a significant amount of sub- 5 \AA micropores ($0.21 \text{ cm}^3/\text{g}$, $400 \text{ m}^2/\text{g}$).

Similarly, by applying a series of probing gases, the pore size of PDA-C800 is analyzed to be $4.1\text{-}4.3 \text{ \AA}$, which falls in-between the molecule sizes of C_3H_6 and C_3H_8 , and thus enables the size-sieving of C_3H_6 and C_3H_8 in breakthrough experiments (Fig. 5b). Also, PDA-C800 possesses a high C_3H_6 capacity of 1.98 mmol/g , due to a significant amount of sub- 5 \AA micropores ($0.18 \text{ cm}^3/\text{g}$, $379 \text{ m}^2/\text{g}$).

3. Looking at this interference of your own data, where on the one hand, almost no favourable sorption property of your carbons is found and the micropores are so well hidden (or only very low amounts of those pores are in the material, which would be my ration explanation in accordance with ref.24), but on the other hand the superior separation values are measured, I find it hard to acknowledge these properties to size sieving effects.

Response: Thanks for the reviewer's comment.

With the above explanations to comment 1 and 2, and corrected misleading writing, the data is consistent. On the one hand, favorable sorption property of our carbons is found (i.e., 2.25 mmol/g of C_2H_4 for PDA-C900) due to the significant amount of sub- 5 \AA micropores (i.e., $0.21 \text{ cm}^3/\text{g}$, $400 \text{ m}^2/\text{g}$). On the other hand, superior separation values are measured and can be ascribed to the size-sieving effect, due to the narrowly distributed pore sizes of these micropores falling right in-between the molecule sizes of olefin/paraffin pairs, analyzed by a series of size-differentiating probing gases in this work.

In addition, it should be mentioned that though the micropores are too small to be probed by N₂ at 77 K, they can be probed by smaller molecules, i.e., CO₂, with reasonably high uptake of 5.60 mmol/g at 195 K and 1.0 bar.

4. This further makes me think about your conclusion: Might this not be a completely different effect? Why do you compare your material to size-sieving MOFs? The introduction should be rewritten, and the focus should lie on size-sieving carbons rather than other materials. The benchmarking is fine, but if this is not a size-sieving effect, maybe a different benchmark needs to be considered.

Response: Thanks for the reviewer's comments and suggestions.

Based on the above clarification, it can be concluded it is a size-sieving effect. The size of pore orifice of PDA-C_x falls in-between the molecule size of olefin and paraffin pairs, which allows the adsorption of smaller olefins and exclude slightly larger paraffin counterparts. Further progresses are making in our lab in this direction, which make us confident with the conclusion as well.

We agree with the reviewer and revised the introduction part by focusing on size-sieving carbons rather than other materials, and just use size-sieving MOFs as the benchmark in comparison. In Page 2-3, line 48-74, the introduction part is revised to ".....The keystone for adsorptive separation is the development of advanced physisorbents with excellent separation performance and scalability¹⁰. Porous carbon materials have demonstrated great prospects and broad application in adsorptive separation due to their rich porosity, excellent stability, and low cost, etc.^{11,12}. Regarding to the olefin/paraffin separation, the conventional equilibrium separation based on favorable enthalpic interaction toward olefins or paraffins cannot confer carbon materials with desirable selectivity due to their similar physicochemical property¹³. In principle, to maximize the separation factor, the ideal physisorbents should have narrower sub- nanometer micropores to match the guest molecule size, thus taking up smaller olefins while completely excluding larger paraffin counterparts by precise regulation of pore sizes or geometries¹⁰. However, the major barrier is the broadly distributed pore size of carbonaceous materials, ranging from sub-nanometer to micro-meter scale, due to the random arrangement of carbonaceous nanodomains with uncontrollable defects¹⁴. Such wide pore size distributions (PSD) inevitably cause the co-adsorption of both olefins and paraffins, and thus poor selectivity at a low uptake ratio^{15,16}.

In recent years, research endeavors have been devoted to design carbon materials with tailorable porosity, especially with favorable ultrahigh surface area and pore volume for gas adsorption and storage^{17,18}, only a few works reported carbons with suitable small micropores matching olefins over the slightly larger paraffin counterparts for selective separation. The carbon of C-CDMOF-2-700 derived from metal organic framework (MOF) were recently reported to enable the separation of C₃H₆ and C₃H₈ via size-sieving effect¹⁹. Still, to our best knowledge, it is rather challenging to tailor the micropores (or micropore orifices) in carbon to a lower size range to distinguish C₂H₄ and C₂H₆ at sub-angstrom precision, meanwhile such small micropores in carbon can be readily tuned to recognize C₃H₆ over C₃H₈ counterpart. Furthermore, the conventional single gas probe technique such as N₂ at 77 K,

and Ar at 87 K mainly detects the larger micropores beyond 5 Å²⁰, thereby the contribution from small micropore below 5 Å in carbons can be often veiled and underestimated. As a complement, multiple gas probe molecules could provide a more comprehensive assessment²¹, with cautiously chosen groups of probing molecules.....”

Please see further explanations on size-sieving effect below:

Adsorptive separation can be categorized into three types: equilibrium separation, kinetic separation, and steric separation (size sieving). Equilibrium separation is based on the thermodynamic affinities between different gas molecules and adsorbents. The reported high-selective adsorbents, typically like CuCl/AC carbon (*Adsorption*, 2016, 22(7), 1013-1022) or Co-MOF-74 (*Angew. Chem. Int. Ed.*, 2012, 124(8), 1893-1896) are utilizing the π -complexation interactions by metal sites (Cu^+ , Co^+ , Ag^+ , etc.) to selectively adsorb olefins over paraffins. Even so, the co-adsorption of a moderate amount of paraffins is inevitable. From the view of material's structure, PDA-Cx has no extra functional sites due to the absence of metal species rather than some heteroatom atoms and π -stacking. Thus, there is no strong enthalpic interaction occurred on the non-polar carbon surface. Our inelastic neutron scattering results further confirmed no strong adsorbent-adsorbate interaction (Page 9, line 226-246). Moreover, PDA-C800 could realize superior separation factor of 36.7 for separating $\text{C}_3\text{H}_6/\text{C}_3\text{H}_8$, but it is almost no separation ability for the same kind $\text{C}_2\text{H}_4/\text{C}_2\text{H}_6$ pair (separation factor of only 1.1). That is a typical feature for size-sieving adsorbents that is only effective for a specific separation system by pore-matching. From the view of gas molecules, the polarizabilities of olefin and paraffin pairs are very similar (C_2H_4 : $42.5 \times 10^{25} \text{ cm}^3$; C_2H_6 : $44.3 \times 10^{25} \text{ cm}^3$; C_3H_6 : $62.6 \times 10^{25} \text{ cm}^3$; C_3H_8 : $62.9 \times 10^{25} \text{ cm}^3$). Thus, in comparison with the other gas pairs like CO_2/N_2 (CO_2 : $29.1 \times 10^{25} \text{ cm}^3$ and N_2 : $17.4 \times 10^{25} \text{ cm}^3$), the separation of olefin/paraffin pairs is very difficult, and the reported equilibrium separation selectivity is lower than 4, but PDA-Cx can realize as high as 24.7 (Fig. 4c), that is only achievable for size-sieving effect. Additionally, kinetic separation is based on the difference of diffusion rate of gas molecules in the pores. According to the kinetic curves (Fig. 5a in the revised manuscript), olefins (C_2H_4 and C_3H_6) could reach the equilibrium within 10 min, while paraffins (C_2H_6 and C_3H_8) show negligible capacity with the increase of time within the whole 100 min, such results ruled out the kinetic effect. Therefore, the size-sieving effect is the most reasonable mechanism that could be considered in this work.

5. Maybe, an easy way to proof the amount of micropores to the total volume of your carbons would be Archimedes density measurements. In gas pycnometer, you would be able to measure densities of your materials outgoing from He and with the correct way of calibration you can go up to SF6. These “skeletal” density could be considered to estimate the volumetric amounts of your micropores.

Response: We appreciate the reviewer's concern with pore volume and suggestion.

Archimedes density measurements can be used to analyze the “skeletal” density, but the compressed density of powders (generally > 20 MPa to reduce the voids between particles) is required for measurements, such high pressure may impact the actual values of small pores. Similar as the reviewer suggested, probing gas adsorption analyzed with Dubinin-Astakhov (D-A) equation is applicable as

referred an effect methodology (*Nat Commun*, **2018**, 9, 3789; *Nat Commun*, **2019**, 10, 4114; *Nat Commun*. **2022**, 13, 1701) to quantify the pore volume and the small pore volume. Accordingly, we applied probing gas adsorption analyzed with Dubinin-Astakhov (D-A) equation using the small probing molecule of CO₂ (molecular size of 3.3 Å) at 273 K to analyze the pore volume above 3.3 Å under the ambient pressure, and N₂ physisorption at 77 K to analyze the pore volume above 5 Å on PDA-C900 and PDA-C800. The data are shown below with decent amount of micropores in-between:

Materials	Pore volume (>3.3 Å) probed by CO ₂ ^a	Pore volume (>5 Å) probed by N ₂ ^b
	(cm ³ /g)	(cm ³ /g)
PDA-C900	0.21	0.008
PDA-C800	0.18	0.003

a: at 273 K; b: at N₂ critical temperature of 77 K

Based on the above results, it can be seen for the two molecular-recognition carbons of PDA-C900 and PDA-C800, the total pore volume above 5 Å is negligible, while the pore volumes above 3.3 Å are 0.21 and 0.18 cm³/g, respectively. Therefore, the major porosity was recognized at sub-5 Å micropore region, which accounts for >96% of the total pore volume.

Reviewer comments, second round review –

Reviewer #2 (Remarks to the Author):

The manuscript with an updated title "Probing Sub-5 Å micropores in Carbon for Precise Light Olefins/paraffins Separation" was greatly improved after revision. All comments were taken into account and appropriate changes were made in the text. Among other things, I appreciate the addition of the results for AC-1 and CMS-1, which give the expected reference for the tested samples. The manuscript is almost ready for publication with very minor corrections. I have only a few comments concerning newly added fragments of the paper:

1. The improvement of the PALS experiment description, data analysis and interpretation of the results involves the use of the relationship between the positron lifetime and free volume size. This is described as: "In our work, the free-volume size was determined by using a semi-empirical infinite potential spherical model" followed by equation 1. This approach is taken from ref. 30 and actually my further discussion concerns rather the interpretation given in ref. 30, which should not be adapted to the reviewed manuscript without any criticism. In fact, using such a model not for neutral positronium but for charged positrons is unjustified. Positron traps are not necessarily the same free volumes that can host adsorbate molecules. Positrons can be trapped in very small closed defects such as vacancies etc.. Moreover, not all positrons are trapped before annihilation and the annihilation rate of this fraction depends on mean electron density of the material. Finally some of these annihilations can originate from quenched positronium. The omission of all these facts leads to the questionable values of the parameters used in the model such as the penetration depth of the positron wavefunction outside the well of 0.38 nm (suggesting very shallow traps and high detrapping probability, which in turn makes model assumptions inaccurate). On the other hand, the existence of such $\tau_2(R)$ relationship is unquestionable and experimental proof can be found in ref. 30. However, purely empirical linear formula given by equation 10 in ref. 30 is much better than the highly questionable attempt to use the model adapted from positronium annihilation. Fortunately changing the equation will not change the results presented in the manuscript, only its description should be reconsidered.
2. The points in Fig. 2c should have uncertainties shown. It is likely that fitting a linear function to these points will give a better impression of the pore size trend than connecting the points with lines.
3. "Maximum full width" is a bit unusual and assumes a zero distribution just outside the peak, but even if it is not completely accurate, it is understandable. Just check that this description is correct. Perhaps this can be extended to avoid confusion.
4. Page 17 and line 456: "equation (1)(2)(3)" should be "equation 2-4" or "equation (2)(3)(4)"
5. In Figure S25, I prefer the previous direction of the x-axis - from small pores on the left to large pores on the right. I understand this change is due to the addition of cumulative pore volume, but this can also be reversed to get the intuitive direction of the axis.

Reviewer #3 (Remarks to the Author):

The manuscript has improved drastically. All the points that were unclear to me have been clarified now. Also, all the points of the other referees have been solved. Many of my questions were based on my misunderstanding of the matter from the earlier version of the manuscript. The authors worked on all my comments with much effort, changed figures and re-edited data tables etc.

from my side, this is now ready for publication. Accept as is.

Response to Reviewers of Manuscript NCOMMS-22-40515-A

We appreciate the Reviewers for their constructive comments and suggestions on our revised manuscript entitled “Probing Sub-5Å micropores in Carbon for Precise Light Olefins/paraffins Separation”. We have carefully considered all the Reviewers’ comments and have revised the manuscript to address their thoughtful concerns. Changes are marked in red in the revised manuscript. A detailed point-by-point response to the reviewers’ comments is attached below.

Reviewer(s)' Comments to Author:

Reviewer: 2

Comments:

The manuscript with an updated title “Probing Sub-5 Å micropores in Carbon for Precise Light Olefins/paraffins Separation” was greatly improved after revision. All comments were taken into account and appropriate changes were made in the text. Among other things, I appreciate the addition of the results for AC-1 and CMS-1, which give the expected reference for the tested samples. The manuscript is almost ready for publication with very minor corrections. I have only a few comments concerning newly added fragments of the paper.

We thank the reviewer for the positive recommendation and constructive suggestions.

1. The improvement of the PALS experiment description, data analysis and interpretation of the results involves the use of the relationship between the positron lifetime and free volume size. This is described as: “In our work, the free-volume size was determined by using a semi-empirical infinite potential spherical model” followed by equation 1. This approach is taken from ref. 30 and actually my further discussion concerns rather the interpretation given in ref. 30, which should not be adapted to the reviewed manuscript without any criticism. In fact, using such a model not for neutral positronium but for charged positrons is unjustified. Positron traps are not necessarily the same free volumes that can host adsorbate molecules. Positrons can be trapped in very small closed defects such as vacancies etc. Moreover, not all positrons are trapped before annihilation and the annihilation rate of this fraction depends on mean electron density of the material. Finally, some of these annihilations can origin from quenched positronium. The omission of all these facts leads to the questionable values of the parameters used in the model such as the penetration depth of the positron wavefunction outside the well of 0.38 nm (suggesting very shallow traps and high detrapping probability, which in turn makes model assumptions inaccurate). On the other hand, the existence of such $\tau_2(R)$ relationship is unquestionable and experimental proof can be found in ref. 30. However, purely empirical linear formula given by equation 10 in ref. 30 is much better than the highly questionable attempt to use the model adapted from positronium annihilation. Fortunately changing the equation will not change the results presented in the manuscript, only its description should be reconsidered.

Response: Thanks for the reviewer’s expert and constructive advice on PALS. We are fully persuaded to accept equation 10 rather than equation 9 in ref. 30 (ref. 38 in the revised manuscript). Accordingly,

we analyzed the average free volume of PDA-Cx materials in the revised manuscript. The experimental description has been improved accordingly. Kindly see below:

Page 13, line 327-332: In this work, the o-Ps intensity (I_3 , formation probability of o-Ps) in each PDA-Cx sample is lower than 1% (Supplementary Table S2). Thus, the average diameter ($2R$) of free-volumes was estimated from τ_2 based on an empirically linear equation, that exhibits a decent correlation coefficient of 0.9268 for τ_2 -R data on polymers, zeolites, and molecular sieves with $R < 5 \text{ \AA}$ ³⁸. The equation is formally written as:

$$\tau_2 = 0.174 (1+0.494R) \quad (1)$$

where the units of τ_2 and R are expressed in ns and \AA , respectively.

Meanwhile, the calculated average size of free-volume according to the equation 10 in the ref. 38 has been corrected in the revised manuscript.

Page 5, line 131-133. "PALS-detected average pore size increased from 4.68 \AA of PDA-C300 to 5.25 \AA of PDA-C900, revealing the formation of slightly larger angstrom-sized voids in carbon matrix induced by higher pyrolysis temperature (Fig. 2c)."

2. The points in Fig. 2c should have uncertainties shown. It is likely that fitting a linear function to these points will give a better impression of the pore size trend than connecting the points with lines.

Response: Thanks for the reviewer's suggestion. Accordingly, the uncertainties of the points in Fig. 2c were added in the revised manuscript. Meanwhile, the points were fitted by a linear function to give a better impression of the average pore size.

Fig. 2c. The average pore size of PDA-Cx derived from the PALS. Dashed line is a linear fit of the data as visual guide.

3. "Maximum full width" is a bit unusual and assumes a zero distribution just outside the peak, but even if it is not completely accurate, it is understandable. Just check that this description is correct. Perhaps this can be extended to avoid confusion.

Response: Thanks for the reviewer's suggestion. To avoid the confusion for readers, we removed the unusual term of "Maximum full width" and corrected the corresponding description in the revised manuscript, accordingly. Kindly see below:

Page 7, line 175-178: "Noticeably, small pore sizes of PDA-C800 and PDA-C900 fell in-between 4.1-4.3 Å and 3.7-4.0 Å, respectively (Fig. 3d). Such narrow size distributions are even comparable with typical crystalline adsorbents such as zeolites and MOFs^{3,4,47,48}, promising great potential for size-sieving separation of olefins/paraffins."

Page 12, line 297-299: "Results show that the small micropore orifices of 4.1-4.3 Å in PDA-C800 and 3.7-4.0 Å in PDA-C900 fall in-between the molecule size of C₃H₆/C₃H₈ and C₂H₄/C₂H₆ pairs, respectively, allowing the entrance of olefins while entirely exclude larger paraffin counterparts."

4. Page 17 and line 456: "equation (1)(2)(3)" should be "equation 2-4" or "equation (2)(3)(4)".

Response: Thanks for reviewer's correction. We corrected the "equation (1)(2)(3)" in Page 17, line 456 (Page 14, line 352 in the revised manuscript) into "equation (2)(3)(4)", accordingly.

5. In Figure S25, I prefer the previous direction of the x-axis - from small pores on the left to large pores on the right. I understand this change is due to the addition of cumulative pore volume, but this can also be reversed to get the intuitive direction of the axis.

Response: Thanks for reviewer's suggestion. Accordingly, we corrected the direction of the x-axis in Figure S25, from small pores on the left to the large pores on the right.

Figure S25. Macroporosity fraction of PDA and PDA-C_x samples obtained from mercury porosimetry.

Reviewer: 3

Comments:

The manuscript has improved drastically. All the points that were unclear to me have been clarified now. Also, all the points of the other referees have been solved. Many of my questions were based on my misunderstanding of the matter from the earlier version of the manuscript. The authors worked on all my comments with much effort, changed figures and re-edited data tables etc. from my side, this is now ready for publication. Accept as is.

Response: We highly appreciate the reviewer's positive comments and recommendation on the revised manuscript for publication.

Reviewer comments, third round review –

Reviewer #2 (Remarks to the Author):

The manuscript "Probing Sub-5 Å micropores in Carbon for Precise Light Olefins/paraffins Separation" is ready for publication. The responses to my comments on the revised version #1 are satisfactory. All corrections made are correct. Congratulations to the authors for a job well done.

Response to Reviewers of Manuscript NCOMMS-22-40515-B

We appreciate the Reviewers for their constructive comments and suggestions on our revised manuscript entitled “Probing Sub-5Å micropores in Carbon for Precise Light Olefins/paraffins Separation”.

Reviewer(s)' Comments to Author:

Reviewer: 2

Comments:

The manuscript "Probing Sub-5 Å micropores in Carbon for Precise Light Olefins/paraffins Separation" is ready for publication. The responses to my comments on the revised version #1 are satisfactory. All corrections made are correct. Congratulations to the authors for a job well done.

Response: We highly appreciate the reviewer’s positive comments and recommendation on the revised manuscript for publication.